# Whole-body visualization of SARS-CoV-2 biodistribution in vivo by immunoPET imaging in non-human primates

Alexandra Detrille[1], Steve Huvelle[1,2], Marit J. van Gils [3,4], Tatiana Geara[1], Quentin Pascal[1], Jonne Snitselaar [3,4], Laetitia Bossevot [1], Mariangela Cavarelli [1], Nathalie Dereuddre-Bosquet [1], Francis Relouzat [1], Vanessa Contreras[1], Catherine Chapon[1], Fabien Caillé[2], Rogier W. Sanders [3,4], Roger Le Grand [1] & Thibaut Naninck [1] ✉

The COVID-19 pandemic has caused at least 780 million cases globally. While available treatments and vaccines have reduced the mortality rate, spread and evolution of the virus are ongoing processes. Despite extensive research, the long-term impact of SARS-CoV-2 infection is still poorly understood and requires further investigation. Routine analysis provides limited access to the tissues of patients, necessitating alternative approaches to investigate viral dissemination in the organism. We address this issue by implementing a whole-body in vivo imaging strategy to longitudinally assess the biodistribution of SARS-CoV-2. We demonstrate in a COVID-19 non-human primate model that a single injection of radiolabeled [89Zr]COVA1-27-DFO human monoclonal antibody targeting a preserved epitope of the SARS-CoV-2 spike protein allows longitudinal tracking of the virus by positron emission tomography with computed tomography (PET/CT). Convalescent animals exhibit a persistent [89Zr]COVA1-27-DFO PET signal in the lungs, as well as in the brain, three months following infection. This imaging approach also allows viral detection in various organs, including the airways and kidneys, of exposed animals during the acute infection phase. Overall, the technology we developed offers a comprehensive assessment of SARS-CoV-2 distribution in vivo and provides a promising approach for the non-invasive study of long-COVID pathophysiology.

The COVID-19 pandemic had caused more than 780 million cases worldwide as of December 2023[1]. This disease is caused by the SARS-CoV-2, a coronaviridae of the same family of viruses responsible for severe acute respiratory syndrome in Asia (SARS, 2003) and Middle East respiratory syndrome (MERS, 2012)[2,3]. Currently available treatments and vaccines[4] have reduced the mortality rate but not the circulation of the virus[1], which is still evolving to escape immune pressure. Despite accumulated knowledge on the pathogenesis of COVID-19 since the discovery of SARS-CoV-2, early dissemination of the virus in tissues, the establishment of a possible viral

[1]Université Paris-Saclay, Inserm, CEA, Center for Immunology of Viral, Auto-immune, Hematological and Bacterial diseases (IMVA-HB/IDMIT/UMRS1184), Fontenay-aux-Roses & Le Kremlin-Bicêtre, Fontenay-aux-Roses, France. [2]Université Paris-Saclay, Inserm, CNRS, CEA, Laboratoire d'Imagerie Biomédicale Multimodale Paris-Saclay, Orsay, France. [3]Department of Medical Microbiology and Infection Prevention of the Amsterdam UMC, University of Amsterdam, Amsterdam, The Netherlands. [4]Amsterdam Institute for Immunology and Infectious Diseases, Amsterdam, The Netherlands. ✉e-mail: thibaut.naninck@cea.fr

reservoir, and the long-term impact of infection (long-COVID) are still not fully understood[5–14] and need to be addressed.

It is well established that the virus mainly localizes to the upper airways, lungs, intestinal tract, and genital tract[15]. Much less is known about dissemination of the virus to other tissues, in particular, the brain. Findings on intra-host viral dissemination and replication are indeed significantly limited, as they are based on samples collected at necropsy from patients who died from severe COVID-19[16,17] or partial sampling of tissues[18,19]. Routine analysis of swab samples by RT-qPCR can only provide limited knowledge of the presence of the virus in the nasopharyngeal cavity, trachea, or rectum[18–20] and does not allow extensive characterization of the dynamics of whole body viral replication and dissemination from initial infection to the onset of disease or possible persistence in the tissues and the establishment of viral reservoirs. A complete understanding of the pathogenesis of COVID-19 still requires new approaches that allow extensive characterization of viral dissemination, persistence, and reservoirs at the whole-body scale[21].

Preclinical models of SARS-CoV-2 infection can help to address these questions. During the last five years, many experimental models of SARS-CoV-2 infection including mice, golden Syrian hamsters, ferrets and non-human primates were developed in order to study COVID-19 pathogenesis, associated immune responses and to test drug or vaccine candidates. Each model has its advantages and limitations, for instance hamsters developed highly severe forms of COVID-19 including cytokine storm which could be used as a model for acute COVID-19 happening in patients in ICU. Ferrets were also described as very useful for transmissibility studies. However, pathophysiology and/or localization of the virus in the airways in these small animal models did not accurately reproduce the human spectrum of the pathology[22]. Among these models, non-human primates (NHPs) are however particularly relevant. They share >93% of their genome with humans and recapitulate many SARS-CoV-2-induced pathogenic mechanisms. As such, macaques possess similar ACE2 cell receptors used by SARS-CoV-2 for infection[23,24]. In most cases, NHPs exhibit mild to moderate symptoms when experimentally infected by SARS-CoV-2[22,25–27]. As in humans, severe symptoms only occur in a few situations and in many cases are associated with comorbidities[28,29]. Over the past five years, NHPs have been widely used to assess the efficacy of treatments (repurposed drugs[25,30,31], new antivirals[30,32], biotherapies[33–35]) and vaccines (inactivated[36–38], live attenuated, mRNA[39–41], DNA[42], recombinant subunits, viral particle derivatives[43–45]).

The cynomolgus macaque (CM) model offers several advantages for studying SARS-CoV-2 infection. It allows control of the inoculated dose and method of exposure, access to daily follow-up of the infection from time zero, and the study of disease progression and host response dynamics. Moreover, the NHP model allows access to tissue analyses required for a better understanding of the biodistribution of the virus in vivo.

Whole body positron emission tomography (PET) imaging is increasingly used in preclinical and clinical research for pharmacokinetic and pharmacodynamic studies, as well as for diagnostics and treatment monitoring[46–50]. Several challenges have thus far limited the extension of this technology to visualizing infection processes and the host response in large animals and humans[21].

Previous studies using [18F]-fluorodeoxyglucose ([18F]FDG) PET with computed tomography (PET/CT) have already shown that SARS-CoV-2 infection in NHPs induces inflammatory lung lesions, together with the activation of secondary lymphoid organs[47], but without evidence of direct viral involvement.

Here, we demonstrate the presence of the SARS-CoV-2 antigens by whole body immunoPET using a radiolabeled non-neutralizing monoclonal antibody that specifically targets a preserved epitope of the spike protein of the virus[51]. We show persistence of the antigen in the lungs and brains of SARS-CoV-2 infected convalescent animals that could be of great interest for long-COVID investigations, as well as the distribution of the virus in various organs of acutely infected CMs.

## Results

### Generation and in vitro validation of the [89Zr]COVA1-27-DFO radiotracer for SARS-CoV-2 detection

We developed a radiolabeled monoclonal antibody (mAb) tracer based on human mAb COVA1-27 for the immunoPET analysis of SARS-CoV-2 infection. This mAb was selected among various other COVA mAb for its high affinity for the SARS-CoV-2 spike protein ($K_D = 0.7$ nM), without being neutralizing. Hence, it was not expected to interfere with viral dissemination[51]. [89Zr]COVA1-27-DFO and an [89Zr]IgG1-DFO negative control were obtained in $86 \pm 4\%$ ($n = 3$) and 73% ($n = 1$) radiochemical yields, respectively, with radiochemical purity >99% and average specific activities of 64 and 27 MBq/mg, respectively (Supplementary Fig. 1).

We evaluated the impact of the radiochemistry on the ability of COVA1-27 to bind SARS-CoV-2 spike protein by comparing the binding of native and DFO-functionalized COVA1-27 to the spike proteins of both the wildtype and Delta variants. DFO functionalization resulted in an decrease in binding of $15.6 \pm 6.3\%$ for the wildtype spike protein and $15.0 \pm 5.6\%$ for the Delta spike protein, without a significant difference ($p > 0.9$) in binding between the two variants (Fig. 1a). Thus, the antibody functionalized with DFO maintained a high affinity for its target, the spike protein.

Subsequently, we assessed the stability of COVA1-27-DFO binding to the Delta spike protein under physiological-like conditions over two weeks. The change in the percentage of binding towards Delta spike with respect to the first day of incubation varied by less than $8.6 \pm 5.7\%$ over time. Thus, the DFO coupled COVA1-27 antibody remained stable in vitro under physiological-like conditions throughout the two-week evaluation period (Fig. 1b).

### In vivo validation of SARS-CoV-2 spike protein detection using the [89Zr]COVA1-27-DFO radiotracer

We first investigated the in vivo recognition of the spike protein by either [89Zr]COVA1-27-DFO or [89Zr]IgG1-DFO. A positive PET signal was observed at the spike protein subcutaneous injection site (left thigh) in animals injected with [89Zr]COVA1-27-DFO on day 3 post injection (D3pi) (Fig. 1c). The PET signal at the spike injection site of animals injected with [89Zr]COVA1-27-DFO was $2.43 \pm 0.78$ times higher than that of animals injected with PBS (right thigh) (Fig. 1d).

Moreover, the signal ratio between the spike and PBS injection sites for one of the [89Zr]IgG1-DFO injected animals (CM8) was close to one, confirming the SARS-CoV-2 spike-specific signal for animals injected with [89Zr]COVA1-27-DFO. The second animal injected with [89Zr]IgG1-DFO showed edema for several days at one injection site and was therefore excluded from this analysis (Fig. 1d, star).

### [89Zr]COVA1-27-DFO accumulates in the lungs and brains of convalescent animals

We evaluated the uptake of the [89Zr]COVA1-27-DFO radiotracer in two convalescent animals (CM1 & CM2) exposed to the SARS-CoV-2 Delta variant three months prior to PET/CT imaging, as well as in two naïve animals (CM3 & CM4, Table 1). The in vivo experimental study timeline is presented in Fig. 2. Following injection with 2.5 mg of the radiotracer, we found no major uptake by the nasal cavity of convalescent animals relative to naïve animals (Fig. 3a–c), in accordance with the absence of detectable viral RNA by RT-qPCR in nasopharyngeal swabs at the same time points. However, we detected [89Zr]COVA1-27-DFO uptake by the lungs of one convalescent animal (CM2) at D7pi of the antibody, along with CT ground-glass opacity (Fig. 3d, e). Tracer uptake by these lesional lung regions (determined by CT) was generally higher than by non-lesional lung areas ($0.28 \pm 0.11$ versus

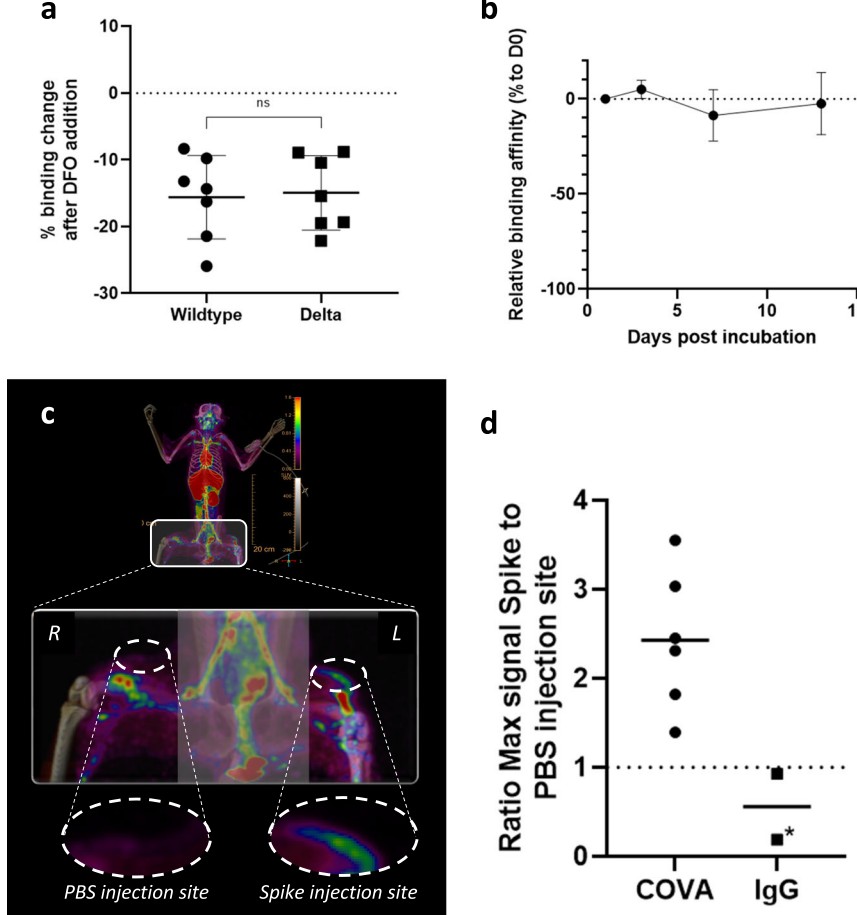

**Fig. 1 | Validation of the COVA1-27-DFO tracer. a** Impact of the addition of DFO on COVA1-27 specificity: Change in binding of native COVA1-27 versus DFO-functionalized COVA1-27 to the spike proteins of the wildtype and Delta variants, presented as the percentage of the ratio of their ECL signals ($n = 7$ samples, unpaired two-sided Mann Whitney $t$-test). **b** Impact of the addition of DFO on COVA1-27 stability: evolution of the binding, expressed as the ratio of binding to that on D0, of COVA1-27-DFO to the Delta variant spike protein over a two-week incubation period in macaque serum at 37 °C ($n = 3$ replicates). **c** In vivo [$^{89}$Zr] COVA1-27-DFO specificity: representative PET/CT image (maximum intensity projection) of the spike (left thigh) and PBS injection sites (right thigh) in the white dashed circles at two days post-injection (p.i.). **d** Associated quantification of PET uptake at 2 dpe: ratio of the maximum signal uptake (SUVmax) at the spike injection site to the that at the PBS injection site for the animals injected with [$^{89}$Zr]COVA1-27-DFO (black circles, $n = 6$) and those injected with [$^{89}$Zr]IgG-DFO (black squares, $n = 2$), star: animal exhibiting edema for several days at the PBS injection site. Data are presented as individual values (**a, d**) with the mean (**a, b, d**) and standard deviation (**a, b**). Source data are provided as a Source data file.

**Table 1 | Group repartition**

| Radiotracer dose | Probe | Exposition ($10^5$ TCDI50) | Follow up | N = | Animal ID | Spike SC injection? |
|---|---|---|---|---|---|---|
| 2,5 mg/animal | COVA1-27 | Convalescent SARS-CoV-2 (B.1.617.2) | 1 week | 2 | CM1, CM2 | Yes |
| | COVA1-27 | PBS | 2 weeks | 2 | CM3, CM4 | Yes |
| | COVA1-27 | SARS-CoV-2 (B.1.617.2) | 2 weeks | 2 | CM5, CM6 | Yes |
| | IgG1 | SARS-CoV-2 (B.1.617.2) | 2 weeks | 2 | CM7, CM8 | Yes |
| | COVA1-27 | SARS-CoV-2 (B.1.617.2) | 3 days | 2 | CM9, CM10 | No |
| 150 µg/animal | COVA1-27 | PBS | 2 weeks | 1 | CM11 | No |
| | COVA1-27 | SARS-CoV-2 (B.1.617.2) | 2 weeks | 2 | CM12, CM13 | No |

$0.21 \pm 0.10$) in these SARS-CoV-2 convalescent animals, without being significant ($p = 0.39$). However, [$^{89}$Zr]COVA1-27-DFO uptake by the persisting lesions was significantly higher than normal lung tracer uptake in naïve animals ($p = 0.008$, Fig. 3f). [$^{89}$Zr]COVA1-27-DFO uptake in non-lesional areas of the lungs of convalescent animals was also significantly higher than in naïve animals as well ($p = 0.014$, Fig. 3f).

Surprisingly, we also found a visible PET signal at D3pi in certain regions of the brains of both convalescent animals (Fig. 3g arrows and Supplementary Fig. 2a, b), with much higher SUV$_{max}$

values than in naïve animals (ratios of 0.80 and 0.63 for CM1 and CM2 versus 0.43 and 0.48 for the CM3 and CM4 controls, Fig. 3h). Moreover, radiotracer accumulation increased from D2pi to D3pi ($2.55 \pm 3.47\%$), whereas the radiotracer was eliminated in the mock + COVA ($-42.62 \pm 10.40\%$) group in the same brain region (Supplementary Fig. 2c). Nevertheless, the average [$^{89}$Zr] COVA1-27-DFO uptake by the brain was similar and extremely low for all animals, indicating only localized uptake in these convalescent animals (Fig. 3i and Supplementary Fig. 2d). Thus,

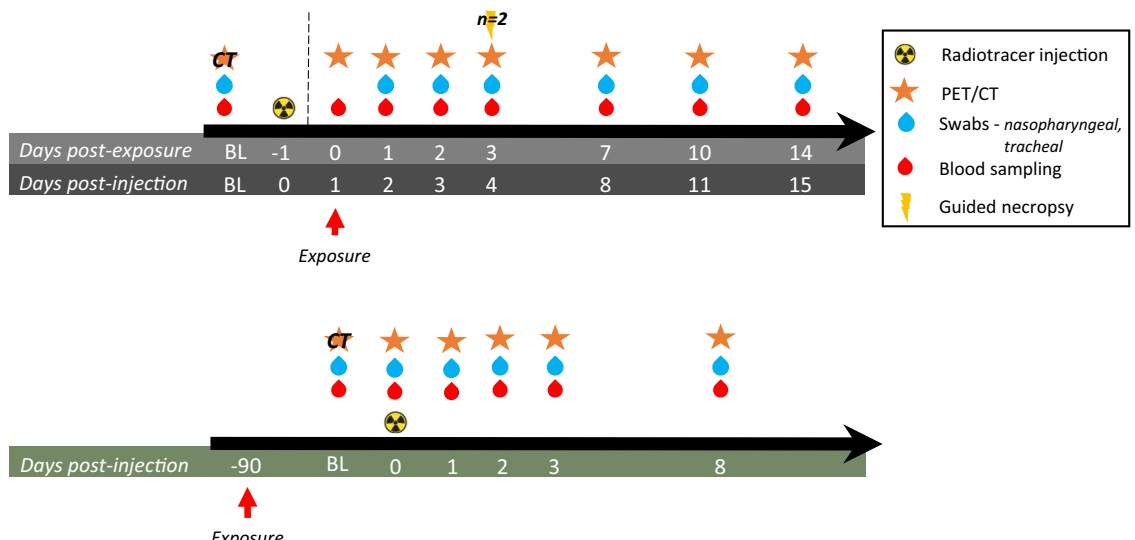

**Fig. 2 | Experimental design.** Design for acute infection (upper panel) investigation or for investigation in convalescent animals (lower panel). PET/CT acquisitions are represented with a star, Swab and blood samples are represented with blue and red droplets respectively.

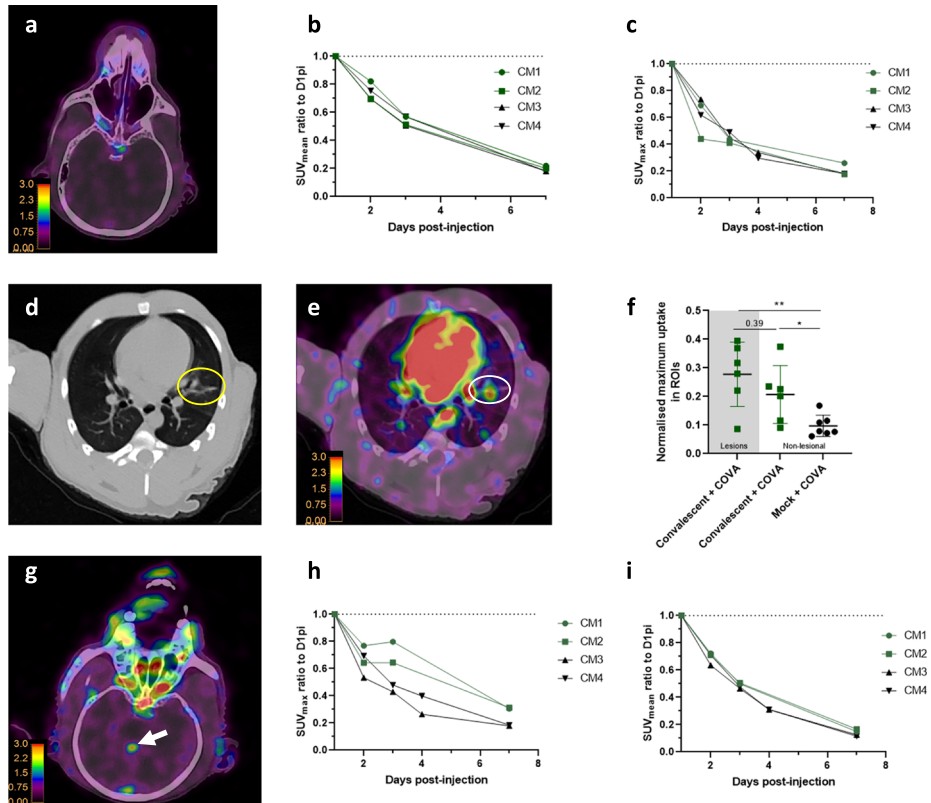

**Fig. 3 | COVA1-27 uptake in convalescent animals. a** Representative PET/CT fusion axial view of the sinus of one convalescent animal (CM2) with the corresponding quantification (**b, c**) of the mean and maximum [⁸⁹Zr]COVA1-27-DFO PET signal in the brain of all animals normalized to that at D1post injection.
**d** Representative CT lung lesion, indicated by the yellow circle, and its associated PET signal (**e**), indicated by the white circle, on D7pi. **f** Corresponding quantification of the maximum PET signal in the lesional (gray background, $n = 6$ areas) and non-lesional areas (square symbols, white background, $n = 6$ areas) on D7pi compared to that for naïve animals (black symbols ($n = 7$ areas). **g** Axial view of the [⁸⁹Zr] COVA1-27-DFO PET signal in the brain, indicated by the white arrows, at D3pi. Corresponding quantification (**h, i**) of the maximum and mean [⁸⁹Zr]COVA1-27-DFO PET signal in the brains of all animals normalized against that on D1pi. Mann Whitney two-sided *t*-test, \*: $p = 0.014$, \*\*: $p = 0.008$. Convalescent + COVA animals are indicated in green and the mock + COVA animals in black. Data are presented as individual values and the associated mean and SD. Source data are provided as a Source data file.

the [⁸⁹Zr]COVA1-27-DFO radiotracer allows detection of the PET signal for SARS-CoV-2 spike antigen up to three months post initial infection in the lungs and brains of COVID-19 convalescent animals.

**[⁸⁹Zr]COVA1-27-DFO injection allows detection of SARS-CoV-2 in the nasal cavity**
To validate our findings in convalescent animals, we then assessed viral detection in a controlled infection NHP model of COVID-19. Four

animals were injected with the [$^{89}$Zr]COVA1-27-DFO radiotracer and then exposed to SARS-CoV-2 (**exposed + COVA group**) and two animals were injected with a negative control [$^{89}$Zr]IgG1-DFO monoclonal antibody radiotracer and then also exposed to SARS-CoV-2 (**exposed + IgG group**). Three other animals ($n = 2$ SARS-CoV-2 infected and $n = 1$ PBS mock-infected) were also injected with a lower dose (150 μg) of the [$^{89}$Zr]COVA1-27-DFO radiotracer. The groups are presented in Table 1. After SARS-CoV-2 ($n = 8$) or mock ($n = 3$) exposure, all animals were monitored for infection via genomic and subgenomic viral RNA (gRNA, sgRNA) detection in nasopharyngeal and tracheal swabs, as well as for clinical conditions, including oximetry, body temperature, and body weight (Supplementary Fig. 3). All animals exposed to the SARS-CoV-2 Delta variant were infected, displaying detectable genomic and subgenomic viral RNA titers in the nasopharyngeal and tracheal cavities (Fig. 4a, b and Supplementary Fig. 4), with no impact of the type of radiotracer injected and without any adverse effects on clinical parameters (oximetry, temperature, and weight, Supplementary Fig. 3). On average, we observed peak infection in exposed animals two days post-exposure (2 dpe) in the nasopharynx, with a viral load (gRNA) of $4.4 \times 10^8 \pm 3.9 \times 10^8$ copies/mL for the [$^{89}$Zr]COVA1-27-DFO-injected CMs and $9.4 \times 10^8 \pm 9.8 \times 10^8$ copies/mL for the [$^{89}$Zr]IgG1-DFO-injected CMs (Fig. 4a, c), confirming the non-neutralizing activity of COVA1-27 in infected animals. In the trachea, we observed peak infection at 2 to 3 dpe during the two week follow-up: reaching $3.1 \times 10^8 \pm 3.8 \times 10^8$ copies/mL, on average, for the [$^{89}$Zr]COVA1-27-DFO injected CMs on D3pe and $1.6 \times 10^8 \pm 1.3 \times 10^6$ copies/mL for the [$^{89}$Zr]IgG1-DFO injected CMs on D2pe (Fig. 4b, d, gRNA). As expected, the [$^{89}$Zr]COVA1-27-DFO-injected, PBS-mock exposed group did not exhibit any infection in either the nasopharynx or trachea (Fig. 4a, b).

All exposed animals showed a similar range for the area under the curve (AUC), both in the nasal (Fig. 4e and Supplementary Fig. 4) and tracheal (Fig. 4f and Supplementary Fig. 4) compartments.

Two of the six animals exposed to SARS-CoV-2 (CM5 and CM7) showed rhinorrhea, with CT opacities in the sinus. In CM5, CT opacities were visible at 3 dpe in the left sinus (Fig. 5a, yellow arrows). They correlated with a positive PET signal at the same location (Fig. 5b, white arrows), as shown in the 3D representation of the nasal cavity (Fig. 5c). In this animal, we detected positive [$^{89}$Zr]COVA1-27-DFO PET uptake starting from 2 dpe and it was also detectable in the other sinus at 10 dpe, along with CT opacity (Supplementary Fig. 5). This pattern was also visible in CM7 following [$^{89}$Zr]IgG1-DFO injection and SARS-CoV-2 exposure but not in the mock + COVA group (Supplementary Fig. 5). The PET signal in the [$^{89}$Zr]COVA1-27-DFO-injected CM5 was 1.4 to 1.6 times higher than that in the [$^{89}$Zr]IgG1-DFO-injected CM7 (Fig. 5d) at 3 dpe and 2dpe, respectively. The other CMs showed a gradual decrease in the PET signal in the nasal cavity over time without any remaining radiotracer uptake (Fig. 5d).

We assessed the specificity of the signal detected by PET imaging by evaluating the evolution of the correlation between SARS-CoV-2 RNA titers and the corresponding radioactivity in the nasal swab samples from 2 dpe (hollow symbols) to 3 dpe (filled symbols) for the three groups. The radioactivity in the nasal swabs increased for both animals in the exposed + COVA group (CM5 & CM6) from 2 to 3 dpe (191% and 41% increase, respectively), with associated increased or stable viral RNA titers (Fig. 5e). On the contrary, the radioactivity decreased in nasal swabs from the two control groups, mock + COVA (CM3, 4) and exposed + IgG (CM7, 8), from 2 to 3 dpe (38% ± 1% and 47% ± 15%, respectively) even with stable RNA titers (CM8) (Fig. 5f). For the mock + COVA group, the viral load was below the limit of detection (Fig. 5e). Overall, these results show that SARS-CoV-2 accumulation in the nasal cavity can be targeted by the [$^{89}$Zr]COVA1-27-DFO probe and detected by PET imaging and gamma counting over time.

## [$^{89}$Zr]COVA1-27-DFO injection allows for SARS-CoV-2 detection in the trachea

As positive RNA titers were detected in the swabs of SARS-CoV-2 exposed animals (Fig. 4b), we investigated [$^{89}$Zr]COVA1-27-DFO and [$^{89}$Zr]IgG1-DFO uptake by the trachea of CMs by PET. We detected qualitative uptake by the trachea of three out of four [$^{89}$Zr]COVA1-27-DFO-injected and infected animals. Localized signals were observed in the tracheal lumen of these animals (Fig. 6a, CM5 at 3 dpe, arrows) but tracheal intubation did not allow us to accurately quantify the signal (Fig. 6b, CM10 at 3 dpe, arrows). Nevertheless, analysis of radioactivity in the tracheal swab samples, as performed for the nasopharyngeal swabs, showed an increase in radioactivity of 146 ± 63% in the tracheal swabs of [$^{89}$Zr]COVA1-27-DFO-injected infected animals from 2 to 3 dpe, with stable viral RNA titers. On the contrary, the two other control groups (exposed + IgG and mock + COVA) showed an 87 ± 2% and 65 ± 3% decrease in tracheal swab radioactivity between 2 and 3 dpe, respectively (Fig. 6c, d). These data suggest that [$^{89}$Zr]COVA1-27-DFO PET imaging in SARS-CoV-2 infected animals enables specific detection of the virus in the trachea during the acute phase of infection.

## [$^{89}$Zr]COVA1-27-DFO injection allows SARS-CoV-2 detection in the lungs

As we observed [$^{89}$Zr]COVA1-27-DFO uptake by the lungs of one of the convalescent monkeys, we investigated CMs for [$^{89}$Zr]COVA1-27-DFO uptake by the lungs following exposure to SARS-CoV-2. Following injection of 2.5 mg of the radiotracer, animals showed visible tracer uptake by the lungs by 7 dpe. As for convalescent CM2, we mainly detected the PET signal at the same location as the CT lung ground-glass opacities, as illustrated in Fig. 7a–d and Supplementary Fig. 6. Lung opacities were either located in the bronchial periphery (bronchopneumonia, Fig. 7a, b) or deeper in the parenchyma (interstitial pneumonia, Fig. 7c, d and Supplementary Fig. 6), as previously described for COVID-19 patients[52]. The CT scores ranged from 1 to 2 for all SARS-CoV-2 exposed animals, whereas the CT score of PBS-mock exposed CMs was 0 (Fig. 7e). Analysis of tracer uptake at 7 dpe in the lesional versus non-lesional regions (determined by CT) showed uptake to be significantly higher in the lesional regions of interest (ROI) than non-lesional ROI of the same animals for the exposed + COVA group ($p = 0.0029$) (Fig. 7f). We observed the same tendency for exposed + IgG animals but without reaching significance ($p = 0.16$), indicating nonspecific IgG1 uptake by these lesions. Tracer uptake by the lesions of exposed animals was even higher relative to that of non-lesional lung parenchyma of PBS-mock exposed animals ($p = 0.0003$ and $p = 0.0007$ for the COVA1-27 & IgG1 tracers, respectively). Comparison of tracer uptake by the lesions of infected animals showed higher uptake of [$^{89}$Zr]COVA1-27-DFO (mean: $0.27 \pm 0.12$) than [$^{89}$Zr]IgG1-DFO (mean: $0.21 \pm 0,06$) but without reaching significance ($p = 0.13$), suggesting effective SARS-CoV-2 detection within these lung lesions. In addition, a comparison of the uptake by non-lesional lung parenchyma for all the groups showed parenchymal tracer uptake to be significantly higher for both the exposed + COVA and exposed + IgG groups than the PBS-mock + COVA group ($p = 0.0052$ and 0.016, respectively). However, the difference in parenchymal uptake between the two SARS-CoV-2-exposed groups was not significant ($p = 0.75$). These data provide evidence of infection-induced but non-specific viral uptake by the lesion-free lung parenchyma of SARS-CoV-2 infected animals (Fig. 7f).

## [$^{89}$Zr]COVA1-27-DFO PET imaging provides insights on SARS-CoV-2 kidney involvement

We also followed [$^{89}$Zr]COVA1-27-DFO and [$^{89}$Zr]IgG1-DFO uptake by various organs of interest over time in SARS-CoV-2-exposed or mock-exposed animals. First, there were no differences in [$^{89}$Zr]COVA1-27-DFO uptake by the brains between SARS-CoV-2-exposed and mock-exposed animals over time, with extremely low signals detected in this

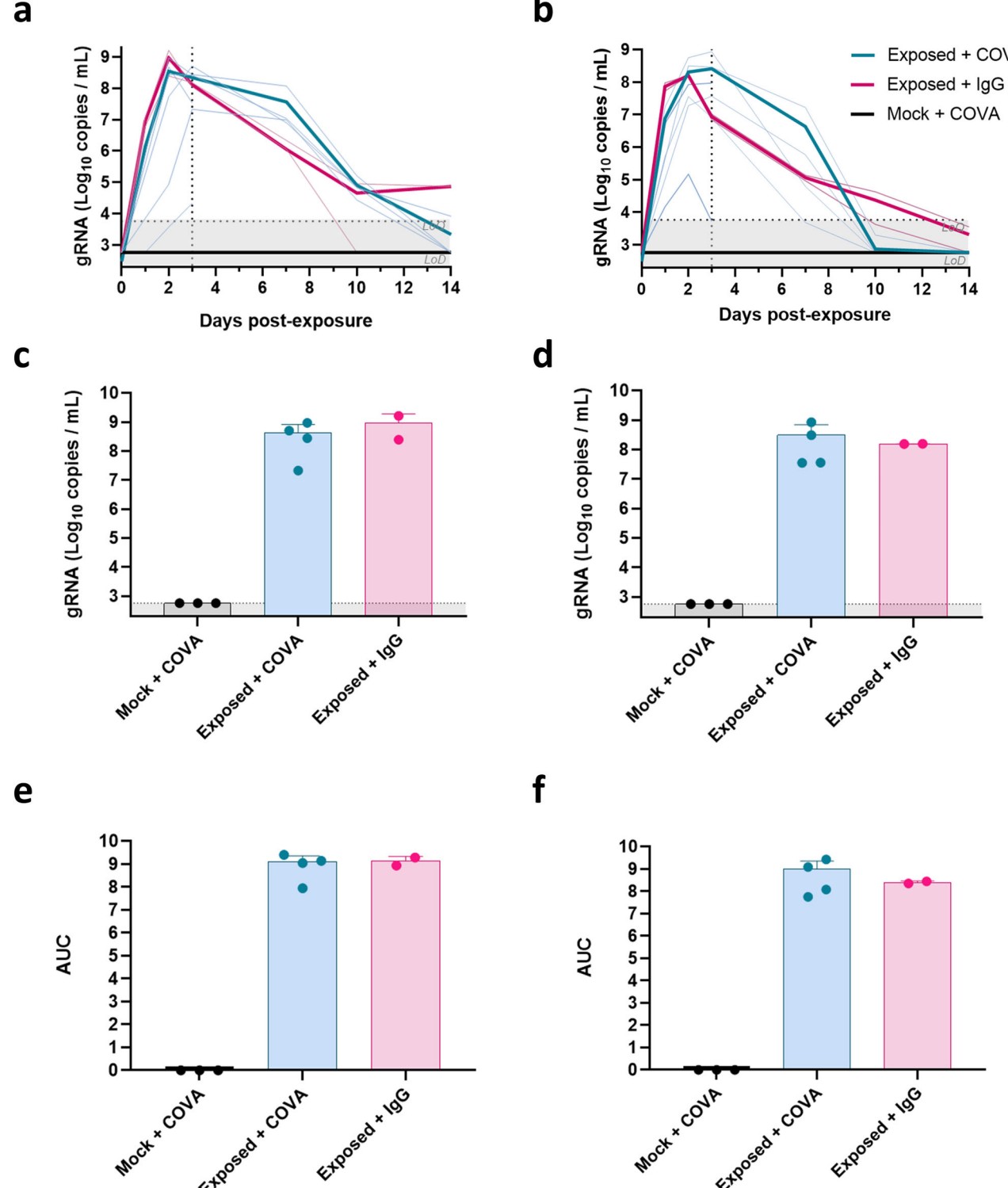

**Fig. 4 | SARS-CoV-2 RNA titers in the nasopharyngeal and tracheal compartments.** Evaluation of the SARS-CoV-2 genomic RNA (gRNA) titers (Log10 copies/mL) over two weeks in the nasopharynx (**a**) and trachea (**b**). The individual follow-up is represented by the thin lines and the average (mean) of each group by the bold lines. Horizontal dotted line: limit of quantification (LoQ)/limit of detection (LoD), vertical dotted line: euthanasia ($n = 2$) during the acute phase of the infection. SARS-CoV-2 gRNA titers (log10 copies/mL) showing the individual peak values for each group in the nasopharynx (**c**) and trachea (**d**). Area under the curve (AUC) calculated for each group in the nasopharynx (**e**) and trachea (**f**). The Mock + COVA group is indicated in black ($n = 3$ animals), the exposed + COVA group in turquoise ($n = 4$ animals), and the exposed + IgG group in pink ($n = 2$ animals). **c**–**f** Data are presented as individual values (circles) with the mean and standard deviation. Source data are provided as a Source data file.

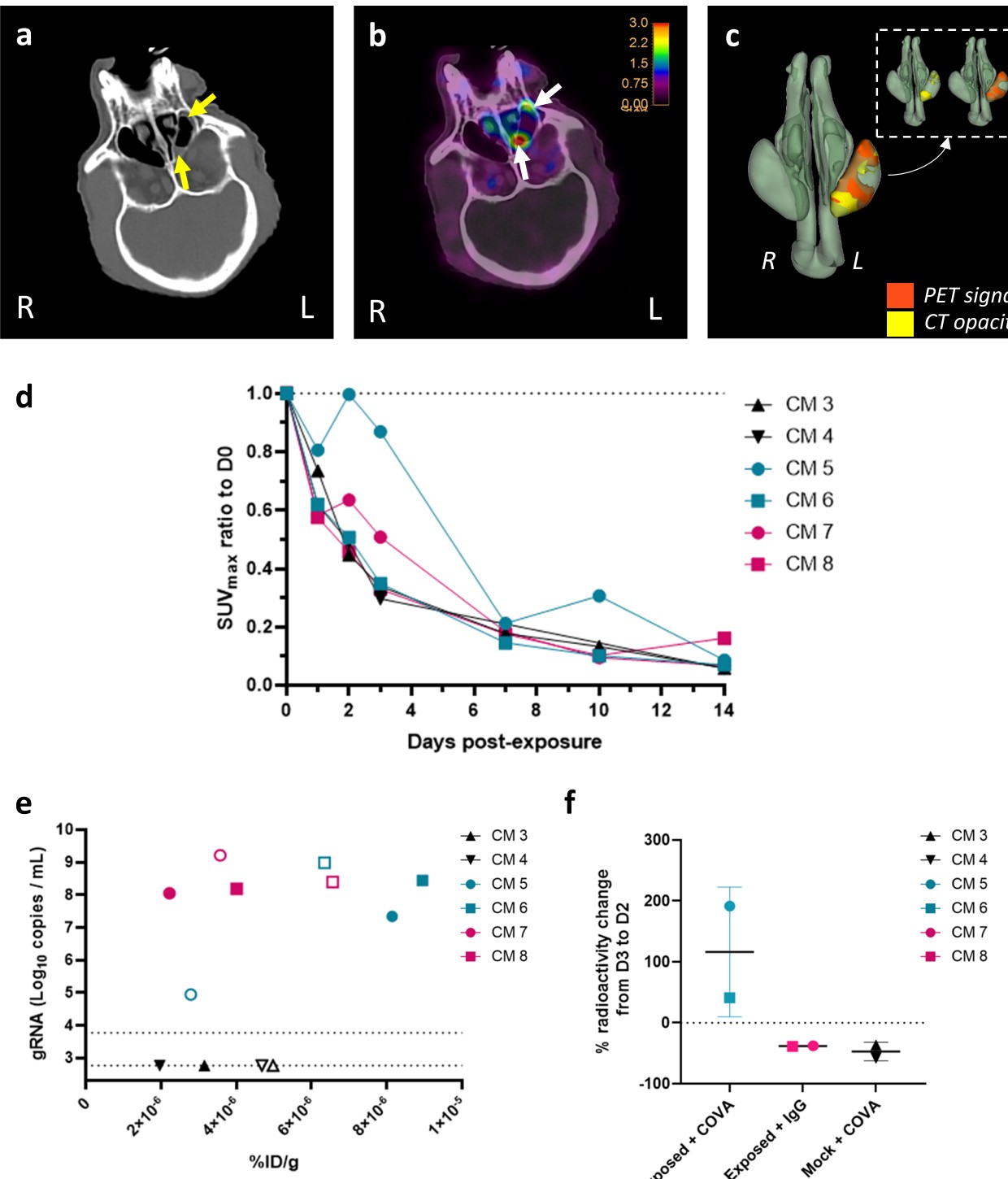

**Fig. 5 | COVA1-27 uptake in the nasal cavity. a** Representative image of CT-detected opacity in the left sinus on day 3 post-exposure (dpe) indicated by the yellow arrows. **b** Corresponding PET signal in the left sinus indicated by the white arrows, color scale: 0–3 SUV. **c** 3D representation of both CT opacity (in yellow) and the PET signal (in orange) in the left sinus of the nasal cavity (in green) at 3 dpe. **d** Longitudinal evaluation of the PET signal (SUVmax) normalized to that at D0 in the nasal cavity. **e** Evolution from 2 (hollow symbol) to 3 dpe (filled symbol) of the radioactivity in the nasopharyngeal swabs correlated with the SARS-CoV-2 gRNA titres in the same sample. **f** Percentage of radioactivity change between 2 and 3 dpe in the nasopharyngeal compartment per group, data are represented with individual values and associated mean values and SD. The exposed + COVA group is indicated in turquoise (circles and squares), the mock + COVA group in black (triangles), and the exposed + IgG group in pink (circles and squares), $n = 2$ animals per group. Source data are provided as a Source data file.

area (Supplementary Fig. 7), consistent with no crossing of the blood brain barrier by the [$^{89}$Zr]COVA1-27-DFO mAb during the acute phase of infection.

However, in other organs, such as the kidneys, we detected [$^{89}$Zr]COVA1-27-DFO and [$^{89}$Zr]IgG1-DFO uptake (Fig. 8a–c, inside the

dotted lines), with progressive elimination of the tracer (Fig. 8d). At 3 dpe, average tracer uptake was 6% higher in infected animals injected with [$^{89}$Zr]COVA1-27-DFO than those injected with [$^{89}$Zr]IgG1-DFO and up to 56% higher than in mock-exposed animals (Fig. 8e).

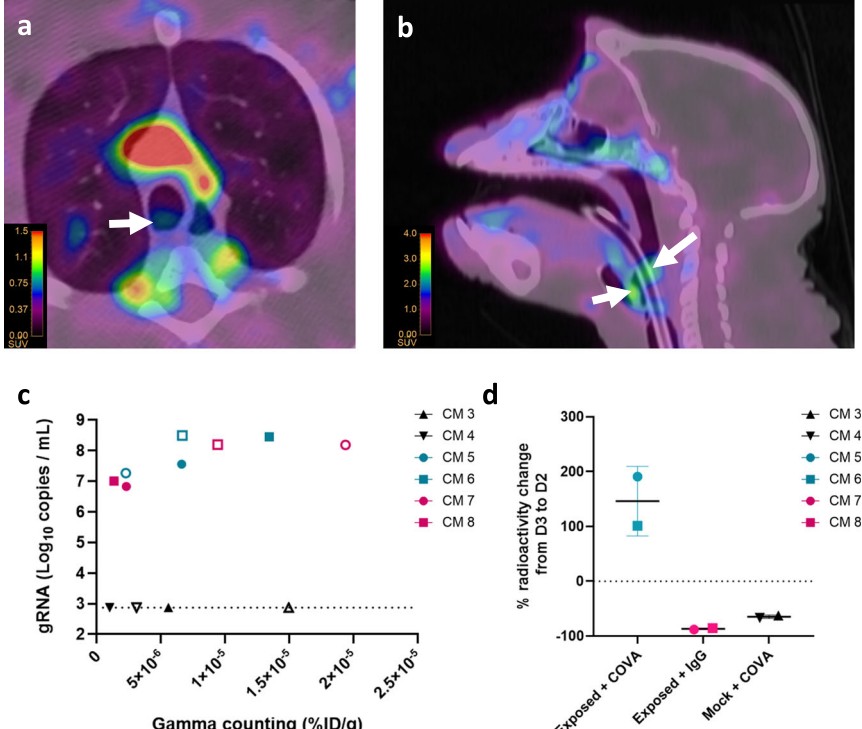

**Fig. 6 | COVA1-27 uptake in the trachea.** Axial (**a**) and sagittal (**b**) views of representative PET/CT images of the COVA1-27 signal in the trachea lumen (arrows) at 3 dpe. **c** Evolution of the radioactivity in tracheal swabs from 2 (empty symbols) to 3 dpe (filled symbols) correlated with the SARS-CoV-2 gRNA titers for the same samples. **d** Evolution of the radioactivity in swabs between 2 and 3 dpe in the

tracheal compartment per group. Data are presented as individual values with the mean and standard deviation. The exposed + COVA group is indicated in turquoise (circles and squares), the mock + COVA group in black (triangles), and the exposed + IgG group in pink (circles and squares), $n = 2$ animals per group. Source data are provided as a Source data file.

## *Postmortem* organ analysis corroborates viral detection by [$^{89}$Zr] COVA1-27-DFO in vivo PET

We performed PET/CT imaging-guided necropsy on two additional animals exposed to SARS-CoV-2 and injected with [$^{89}$Zr]COVA1-27-DFO (CM9 & CM10) to validate the in vivo PET signal and sample analysis data. Various PBS-rinsed organs were placed on the PET acquisition table to more precisely locate the [$^{89}$Zr]COVA1-27-DFO PET signal and reduce blood-induced background noise. The lung lobes of one representative animal are shown in Fig. 9a. Tissue sections for gamma counting, RT-qPCR SARS-CoV-2 RNA quantification, and in situ hybridization were also taken according to the detected PET signal. There was a significant positive correlation between tissue SARS-CoV-2 RNA titers and the radioactivity in the pooled tissue of the same organs of interest (excluding the liver, listed in Supplementary Table 1) ($R = 0.68$, $p = 0.0001$ (Fig. 9b)). In addition, in situ hybridization of tissue from the lungs of these animals more precisely confirmed the distribution of the virus in the pulmonary bronchi and parenchyma (Fig. 9c and Supplementary Fig. 8, **CM10**), as previously detected by CT. Corresponding hematoxylin eosin (HE) staining also allowed histological characterization of the viral distribution. The signal was mainly located in the prismatic ciliated bronchial epithelium (Fig. 9c), alveolar macrophages, and type II pneumocytes (Supplementary Fig. 8), as already reported[53–56]. Furthermore, we validated [$^{89}$Zr]COVA1-27-DFO PET uptake by the kidneys of exposed animals by positive RT-qPCR SARS-CoV-2 RNA quantification following necropsy. CM10, which showed sevenfold higher RNA titers in the kidneys (20,857 copies/250 ng total RNA) than CM9 (2909 copies/250 ng total RNA) had a kidney stone, detected by CT (Fig. 9d, arrow), associated with local acute [$^{89}$Zr] COVA1-27-DFO PET uptake (Fig. 9e, arrow). SARS-CoV-2 RNA was detected in the kidney collecting tubules (brown spots in Fig. 9f) of this animal in the same region. Overall, the *postmortem* tissue

analysis corroborated the viral distribution in these organs measured by [$^{89}$Zr]COVA1-27-DFO PET uptake.

## Low-dose [$^{89}$Zr]COVA1-27-DFO injection does not allow accurate viral detection by PET

To reduce the potential biological effects of the [$^{89}$Zr]COVA1-27-DFO radiotracer in NHPs, we reproduced the same imaging protocol using a lower dose of antibody (150 µg per animal, CM11, 12, and 13). However, we detected no signals above background in the upper or lower airways of the animals by PET imaging, regardless of the time of acquisition. There was also no increase in the uptake of the radiotracer in the lesional areas of the lungs, as detected by CT following exposure (Supplementary Fig. 9a, b, arrows). Quantitative analysis after total lung segmentation showed a gradual decrease in the SUV$_{mean}$ and SUV$_{max}$ values (normalized against those of D0 for each animal) for lung uptake for all CMs (Supplementary Fig. 9c, d). We observed a similar trend for the nasopharyngeal cavity for the SUV$_{max}$ values (Supplementary Fig. 9e). In addition, there was little correlation between viral load (gRNA, Log$_{10}$ copies/mL) and the associated gamma count (%ID/g) in nasal swab samples at 2 dpe (Supplementary Fig. 9f, hollow symbols) and 3 dpe (Supplementary Fig. 9f, filled symbols). For CM13, we observed a decrease in radioactivity ($9.81 \times 10^{-6}$ to $7.18 \times 10^{-6}$% ID/g) in a sample with a stable viral load ($2.77 \times 10^{8}$ to $2.64 \times 10^{8}$ RNA copies/mL) from 2 dpe to 3 dpe. For the other SARS-CoV-2 exposed animal (CM12), both the radioactivity ($6.25 \times 10^{-6}$ to $1.10 \times 10^{-5}$% ID/g) and the viral load ($5.43 \times 10^{7}$ to $5.09 \times 10^{8}$ copies/mL) increased from 2 dpe to 3 dpe, indicating possible virus-induced uptake. The viral load was below the limit of detection for the mock-PBS exposed, [$^{89}$Zr]COVA1-27-DFO-injected CM11 and the radioactivity decreased from $5.17 \times 10^{-6}$ to $2.15 \times 10^{-6}$ % ID/g from 2 dpe to 3 dpe, showing the natural tracer elimination over time. Overall, these low-dose results indicate that a higher tracer concentration is needed to

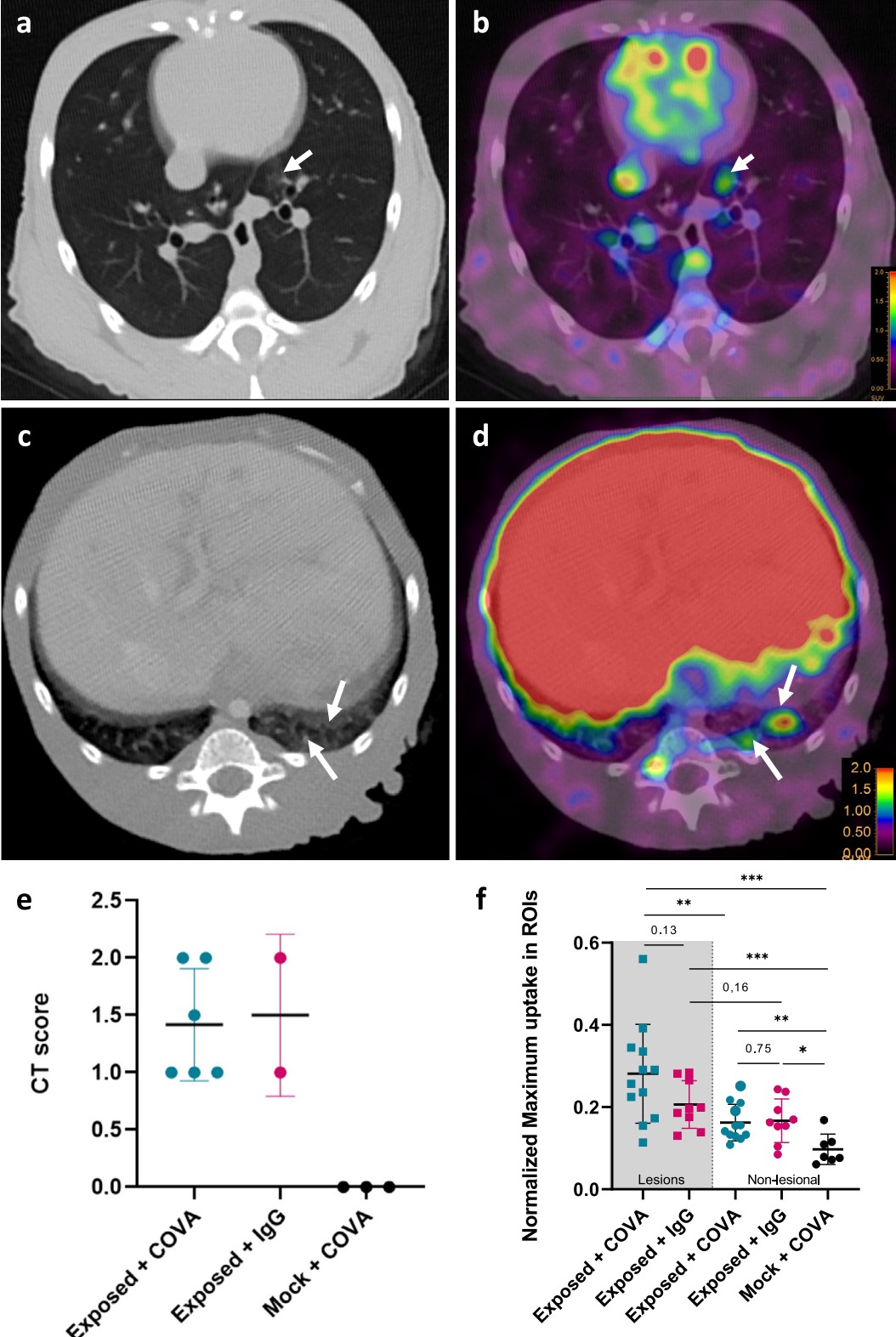

**Fig. 7 | COVA1-27 uptake in the lungs. a**, **c** Representative ground-glass opacity CT lung lesions indicated by the white arrow and (**b**, **d**) the associated PET signal (arrows) at 7 dpe. **e** CT score grading of lung lesion severity at 2 dpe. Data are presented as individual values with the mean and standard deviation ($n = 6$ animals in exposed + COVA group, $n = 2$ animals in exposed + IgG group, $n = 3$ animals in mock + COVA group). **f** Quantification of the maximum PET signal in the lesional (squares, gray background) and non-lesional areas (circles, white background) of the lungs of animals ($n = 12$ ROI in exposed + COVA group, $n = 9$ ROI in exposed + IgG group and $n = 7$ ROI for mock + COVA group). Data are presented as individual ROI values with the mean and standard deviation. Mann Whitney two-sided $t$-tests, *$p < 0.05$, **$p < 0.01$, ***$p < 0.001$. The exposed + COVA group is indicated in turquoise, the exposed + IgG group in pink and, the mock + COVA group in black. Source data are provided as a Source data file.

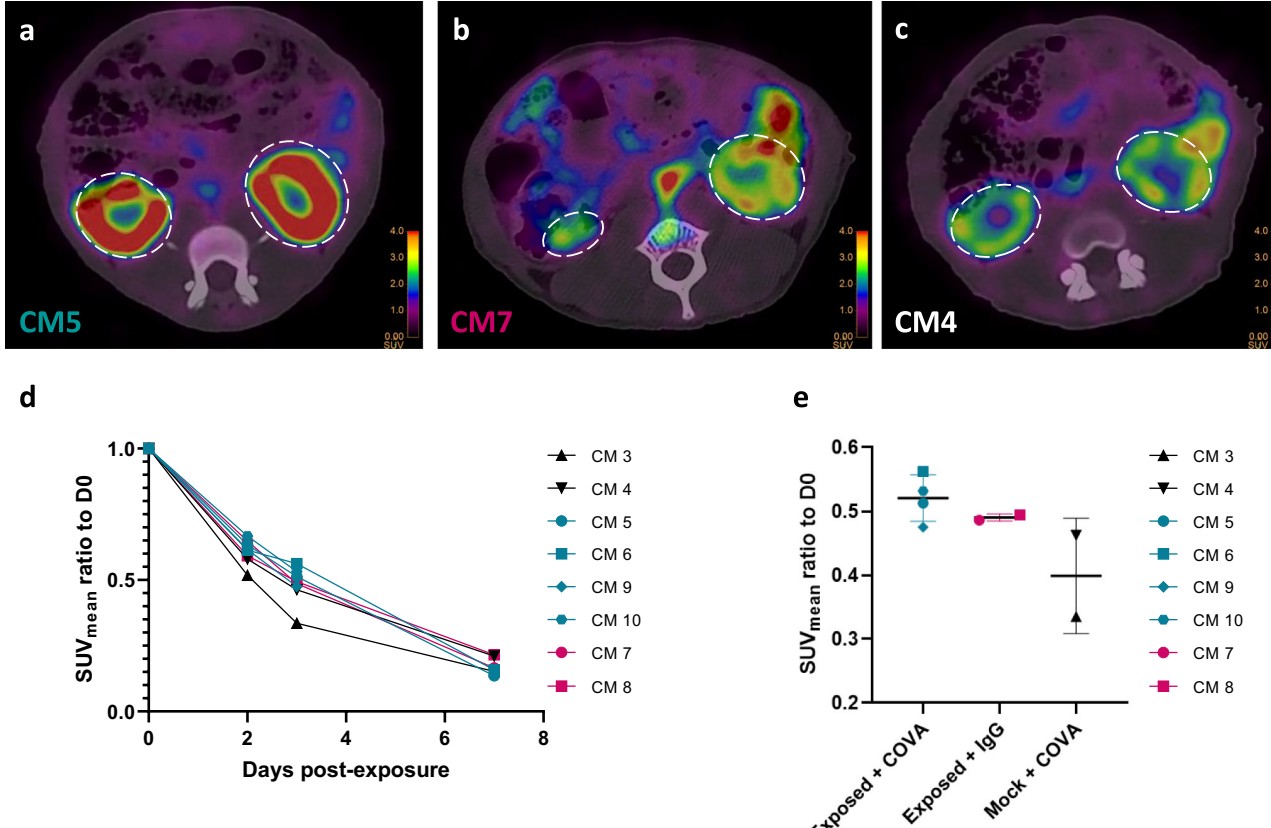

**Fig. 8 | COVA1-27 tracer uptake in the kidneys.** Representative PET signal in the kidneys of CM4 (**a**), CM8 (**b**), and CM7 (**c**), color scale 0–4 SUV. **d** Longitudinal evaluation of the PET signal SUVmean normalized against that of 0 dpe in the kidneys of animals and **e** quantification of the mean [$^{89}$Zr]COVA1-27-DFO PET signal normalized to that of 0 dpe in the kidneys. The exposed + COVA group is indicated in turquoise (circles, squares, hexagons, diamonds, $n = 4$), the exposed + IgG group in pink (circles, squares, $n = 2$), and the mock + COVA group in black (triangles, $n = 2$). Data are presented as individual values the the mean and SD. Source data are provided as a Source data file.

achieve efficient SARS-CoV-2 detection by PET/CT and gamma counting in this animal model of COVID-19.

## Discussion

Despite extensive research efforts since the onset of the COVID-19 pandemic concerning the pathophysiology of the disease and the development of prophylactic or therapeutic strategies, many issues are yet to be addressed. The continuously evolving nature of COVID-19, with emerging new variants of interest, necessitates ongoing research to better understand both COVID-19 and long-COVID patterns. This study aimed to develop an imaging strategy that allows the longitudinal tracking of the biodistribution of SARS-CoV-2 at the whole-body scale in a NHP model of COVID-19 in order to better understand SARS-CoV-2 dissemination in various organs right after exposure but also to assess potential viral antigen persistence in time that could explain the chronic inflammation observed in long-COVID patients. CMs have been established as a relevant model of infection, especially for reproducing the mild to moderate symptoms of COVID-19[25,27]. We applied an antibody-based immunoPET imaging process in a CM model of COVID-19 to assess potential viral antigen reservoirs and longitudinally track the virus at the whole-body scale.

We used a radiotracer molecule capable of targeting the spike protein of SARS-CoV-2 without neutralizing it, the [$^{89}$Zr]COVA1-27-DFO monoclonal antibody (mAb). The original COVA1-27 antibody was initially isolated from a COVID-19 patient and exhibited nanomolar affinity to SARS-CoV-2 wildtype spike protein without neutralizing activity[51]. We functionalized this mAb with DFO and coupled it to radioactive Zirconium-89 to obtain the [$^{89}$Zr]COVA1-27-DFO mAb. We

then assessed the impact of these modifications on spike recognition both in vitro and in vivo. As expected, COVA1-27 functionalization with a DFO chelator resulted in a small decrease in spike-protein binding in vitro but did not significantly affect the initial mAb affinity. Interestingly, COVA1-27-DFO recognized spike proteins from different SARS-CoV-2 variants, including the Delta variant, and showed stable recognition over two weeks under physiological-like in vitro conditions, making it a suitable candidate for a longitudinal follow-up in vivo. In addition, the subcutaneously injected spike protein was specifically recognized in all animals injected with [$^{89}$Zr]COVA1-27-DFO, but not by the negative control nonspecific [$^{89}$Zr]IgG1-DFO, supporting the radiotracer's specificity in vivo towards SARS-CoV-2 spike protein and making it an excellent candidate as a PET imaging probe to track the virus in vivo.

Our initial data on convalescent animals showed [$^{89}$Zr]COVA1-27-DFO uptake to be detectable in persisting lung lesions of one convalescent animal (3 months after exposure), consistent with the results of a previous study showing the persistence of SARS-CoV-2 in the lungs for up to 18 months after infection[57]. We also detected local accumulation of the radiotracer in areas of the brain of both convalescent animals from D2pi to D3pi, consistent with previous findings of neuroinflammation in humans infected with SARS-CoV-2[58] and in rhesus macaques up to six weeks after SARS-CoV-2 infection[59]. The localized crossing of the blood-brain barrier (BBB) by the radiotracer in convalescent animals can be explained by thrombo-inflammation previously reported in patients with active long-COVID[60].

To confirm the specificity of these data in convalescent animals, we then assessed detection of the virus in a controlled infection model

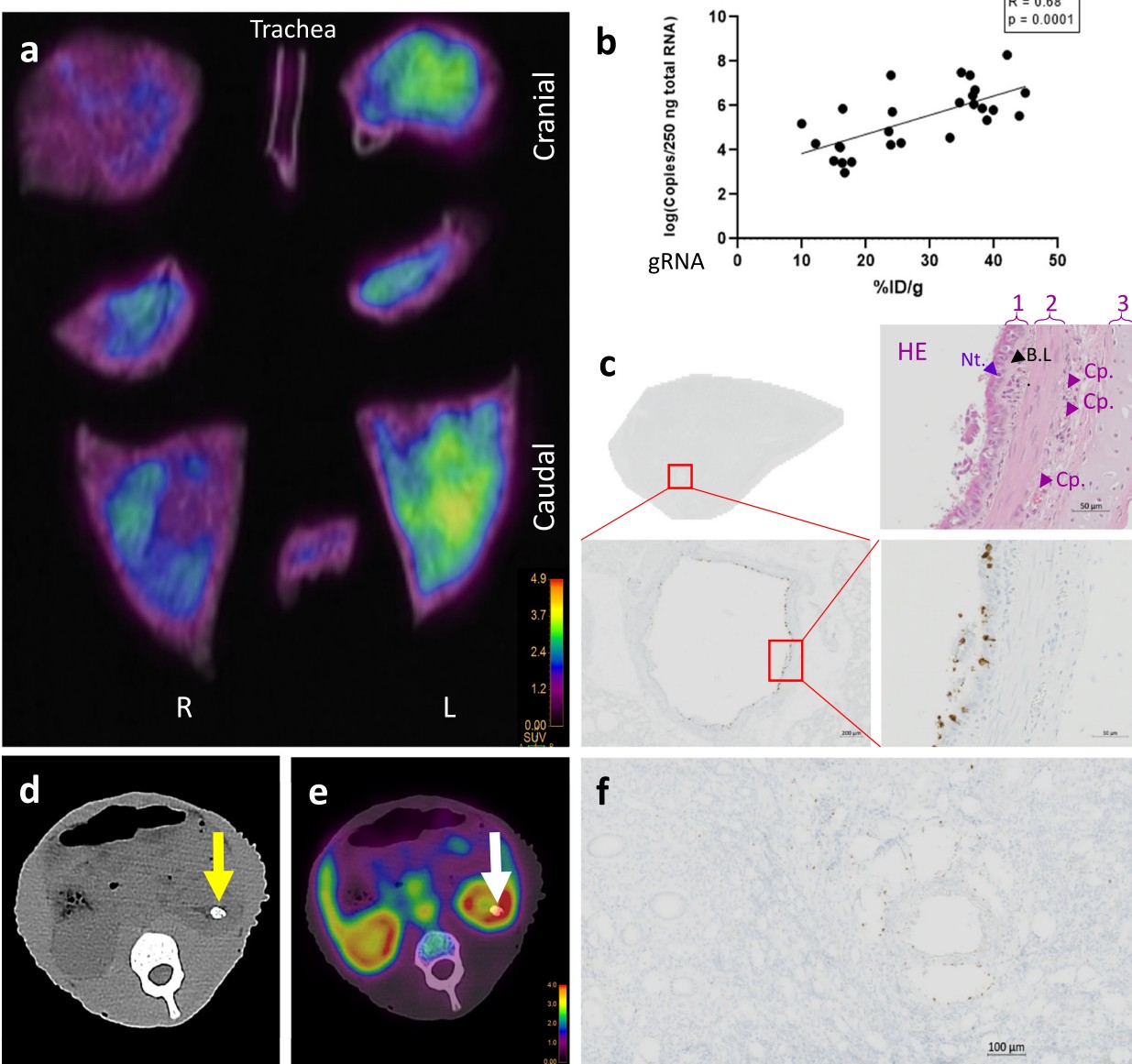

**Fig. 9 | *Postmortem* analysis of the organs. a** PET/CT representative coronal slice of the isolated rinsed lung lobes and trachea *postmortem*. **b** Linear regression showing the correlation between the SARS-CoV-2 gRNA titers in all harvested organs and their associated radioactivity (expressed as %ID/g). **c** Tissue- and cell-scale evaluation of the distribution of the virus (brown signal) by in situ hybridization (ISH) in the lungs (caudal left lobe), with a zoom of a pulmonary bronchus (red square) and the corresponding histological characterization by hematoxylin eosin (HE) staining. Cp capillary, B.L. basal lamina, Nt. neutrophil. 1. Prismatic ciliated bronchial epithelium, 2. Muscle layer, and 3. Cartilage. Representative image of findings in the two animals. **d** CT of kidney stone indicated by the yellow arrow, **e** associated PET signal (white arrow) at 3 dpe, and **f** cellular-scale evaluation of the distribution of the virus (brown signal) by ISH in the left kidney of CM10 *postmortem*. Source data are provided as a Source data file.

of COVID-19 in NHPs. Following exposure to the SARS-CoV-2 Delta variant, CMs exhibited classical mild pathology, including nasopharyngeal and tracheal infection and pulmonary ground-glass opacity lesions, without a major impact on clinical parameters (body temperature, oximetry, weight). Using the aforementioned radiotracer in infected animals, we were able to successfully detect a [$^{89}$Zr]COVA1-27-DFO PET signal in the nasal cavity, trachea, and lungs during the acute phase of the disease. Interestingly, the kinetics and localization of SARS-CoV-2 shedding in the nasal cavity could be addressed in certain animals by [$^{89}$Zr]COVA1-27-DFO PET-CT imaging. Indeed, we detected the virus passing from one sinus at the peak of infection to the other a few days later. Moreover, we observed a localized signal in the tracheal lumen of exposed [$^{89}$Zr]COVA1-27-DFO-injected animals, indicating the presence of the virus. These imaging observations were corroborated by analysis of nasopharyngeal and tracheal swab samples using RT-qPCR and gamma counting, showing that the radiotracer was able to specifically detect SARS-CoV-2 accumulation over time, in agreement with viral RNA titers in these samples. As the macaques exhibited heterogeneous symptoms, reflecting disease variability in patients, we did not observe a PET signal in the sinus of all exposed animals. In addition, robust tracheal quantification of the observed PET signal could not be performed due to tracheal intubation of the animals for imaging. A larger cohort of animals would be needed to quantitatively and statistically confirm these findings.

In the lower respiratory tract of SARS-CoV-2 exposed animals, we found that the areas of lung lesions showed significantly higher uptake of [$^{89}$Zr]COVA1-27-DFO by PET than any other non-lesional areas in all animals, including those that were not infected. Such [$^{89}$Zr]COVA1-27-

DFO uptake was even higher than [$^{89}$Zr]IgG1-DFO uptake by the lesional areas of the lungs, but without reaching significance, suggesting potential nonspecific uptake of these mAb radiotracers through the Fc-gamma receptors of immune-cell infiltrates in the lesions, in addition to virus-specific targeting. This issue could be addressed in future experiments by refining negative controls using potential alternative monoclonal non-specific IgG as negative control or by using antibody fragments (for example, F(ab)'2 or nanobodies) instead of whole IgG as tracers. However, such mAb fragments generally have a much shorter biological half-life, which could limit longitudinal follow-up of the distribution of the virus.

In addition, we detected greater [$^{89}$Zr]COVA1-27-DFO uptake by the kidneys of infected animals than those of control animals, consistent with previous studies indicating the detection of SARS-CoV-2 detection in the kidneys of COVID-19 patients[61].

To confirm our imaging findings in the lungs and kidneys, we performed PET-CT guided necropsy at the peak of infection on two exposed animals. Tissue analysis of the lung and kidney tissue confirmed the presence of viral RNA by RT-qPCR and in-situ hybridization at the same location as the detected [$^{89}$Zr]COVA1-27-DFO PET signal, strengthening our results.

We used a limited number of animals per experimental condition, reducing the possibility of obtaining statistically significant results. However, our findings are supported by the inclusion of several control conditions (mock-exposed animals or use of nonspecific radiotracer) and several types of sample analysis. These imaging results could be statistically confirmed with a larger animal cohort in the future.

Further COVA1-27 immunoPET imaging studies could also be conducted on animals exposed to other SARS-CoV-2 variants of interest or using other exposure routes (e.g., aerosols), as these conditions may better reproduce the natural infection[62]. To better investigate the presence of the virus in the brain in the future, a nanobody-based radiotracer could be used to increase BBB penetration. Autoradiography and histological or RT-qPCR analysis of tissues could also confirm our imaging findings in convalescent animals.

In this study, we were able to directly detect SARS-CoV-2 antigens in vivo using a non-invasive imaging technique. This approach, combining direct virus targeting and longitudinal follow-up, provides a powerful innovative tool to study COVID-19 pathophysiology and viral spreading. It can also provide insights for long-COVID studies by studying antigen persistence in key organs such as brain and lungs, as well as for potential treatment and vaccine efficacy studies in the future by assessing their impact on SARS-CoV-2 distribution in tissues.

## Methods
### Virus
The SARS-CoV-2 Delta (B.1.617.2) variant was purchased from the BEI Resources Repository (Ref NR-55612, Batch 70045240, National Instituted of Health, USA) and was stored at −135 °C until administration. The Delta strain SARS-CoV-2, hCoV-19/USA/PHC658/2021 (B.1.617.2) was produced in Calu-3 cells using two passages: titer = $6.45 \times 10^5$ TCID$_{50}$/mL, volume = 1 mL.

### Description, evaluation, and radiochemistry of the radiotracer
**Production of COVA1-27.** The COVA1-27 human IgG1 monoclonal antibody was previously described[51] and generated as follows. A suspension of HEK293F cells (Invitrogen, cat no. R79007) was cultured in FreeStyle medium (Gibco) and co-transfected at a density of 0.8–1.2 million cells/mL with a 1:1 ratio of the IgG heavy and light chain plasmids of COVA1-27 together with 1 mg/L PEImax (Polysciences) at a 1:3 ratio. After five days, the COVA1-27 antibody was purified by centrifugation of the cell supernatant for 25 min at 4000 rpm, filtered using 0.22-μm pore size SteriTop filters (Millipore), and run over a

10 mL protein A/G column (Pierce), followed by two column volumes of PBS wash. COVA1-27 was eluted from the column using 0.1 M glycine pH 2.5 into neutralization buffer (1 M TRIS pH 8.7) at a 1:9 ratio. Finally, the buffer was changed to PBS using 100 kDa VivaSpin20 columns (Sartorius). The IgG concentration was determined using a NanoDrop 2000 and the antibodies were stored at 4 °C.

**Chemicals and molecules.** A nonspecific whole human IgG1 kappa molecule control was purchased from Thermo Fisher Scientific (No. 31154, 11.6 mg/mL, purified in PBS). [$^{89}$Zr] (185–370 MBq, 1.0 M oxalic acid solution) was produced by BV Cyclotron (The Netherlands) and purchased from Reevity (France).

**General method for size-exclusion chromatography (SEC).** SEC was performed using an Alliance e2695 system equipped with a 2489 UV/Vis detector (Waters, USA) and a HERM LB 500 with an fLumo (Berthold, France) gamma detector. The system was operated using Empower® 3 (Waters, USA) software. SEC was performed on a Superdex 200 Increase 10/300GL (Cytivia, USA) column using PBS (0.1 M, 0.5 mL/min) as eluent. UV detection was performed at 280 nm. The chemical identification was carried out by comparing the retention time of the radiolabeled antibody with that of the non-radioactive antibody without chemical modification (tRref). Radiochemical purity was calculated as the ratio of the area under the curve (AUC) of the radiotracer peak over the sum of the AUCs of all other peaks on gamma chromatograms. The radiochemical yield was calculated as the ratio of the activity of the radiolabeled antibody at the end of the synthesis, measured in an ionization chamber (Capintec®, Berthold, France), over the starting activity afforded to the Zr-89. Specific activity was calculated as the ratio of the activity of the collected peak of [$^{89}$Zr]Ab measured in an ionization chamber (Capintec®, Berthold) over the mass of the antibody determined using a calibration curve.

**Antibody functionalization with p-SCN-Bn-DFO.** COVA1-27 and non-specific IgG1 kappa antibodies were functionalized with 1-(4-iso-thiocyanatophenyl)-3-[6,17-dihydroxy-7,10,18,21-tetraoxo-27-(N-acetylhydroxylamino)-6,11,17, 22-tetraazaheptaeicosine] thiourea (p-SCN-Bn-DFO, simplified as DFO in this study). COVA1-27 and non-specific IgG1 kappa antibody (6–10 mg) solutions (in 0.1 M PBS) were concentrated by centrifugation (Vivaspin, 50 kDa) until the final volume was <1 mL. The final concentration was determined using a Nanodrop and was typically between 5 and 10 mg/mL. The volume was adjusted to 1 mL using a 0.9% NaCl solution. The pH was measured using a 780 pH-meter (Metrohm, France) and adjusted to 9.1–9.3 using a 0.1 M Na$_2$CO$_3$ solution. A 10 mM p-SCN-Bn-DFO solution in DMSO (20 μL, 5.0 equiv. of p-SCN-Bn-DFO relative to the antibody) was added stepwise (5 μL steps) to the solution of antibody while shaking. The resulting mixture was stirred at 500 rpm using a Thermoshaker (ThermoFischer Scientific, USA) at 37 °C for 1 h. At the same time, a PD-10 column (GE Healthcare, USA) was rinsed with 20 mL 0.1 M PBS. The reaction mixture was loaded onto the column and the flow-through discarded. An additional 1.5 mL of PBS 0.1 M was loaded and the flow-through again discarded. Finally, 2 mL 0.1 M PBS was added and the eluted solution was collected. The resulting functionalized antibody was analyzed by SEC following the general method.

**Radiolabeling of functionalized antibodies with Zr-89.** The functionalized COVA1-27 and non-specific IgG1 kappa antibodies were radiolabeled with Zr-89 according to the protocol of Vosjan et al. [63]. First 200 μL (150-200 MBq) of the [$^{89}$Zr]oxalate solution in 1 M oxalic acid buffer was transferred to an Eppendorf tube and Na$_2$CO$_3$ (2 M, 90 μL) was added. The solution was shaken for 3 min and the pH measured and had to be approximately 8. An aqueous solution of HEPES (0.5 M, 710 μL) was added, followed by the addition of 1 mL of the functionalized antibody solution. The resulting mixture was

stirred at room temperature for 1 h. At the same time, a PD-10 column (GE Healthcare, USA) was conditioned with 20 mL of a solution of gentisic acid (5 mg/mL) in 0.25 M sodium acetate buffer. The crude reaction mixture was loaded onto the column and the gentisic acid solution used as eluent. The flow-through was collected in 500-μL fractions. The radioactivity of the collected fractions was measured using a Capintec® device (Berthold, France) and the fractions showing the highest activity (typically fractions 4–7) were pooled and analyzed by SEC following the general method.

**In vitro assessment of COVA1-27 binding and stability.** The binding affinity and stability of the functionalized COVA1-27-DFO antibody to the wildtype (ancestral lineage A) and Delta (lineage B.1.617.2) spike proteins were assessed in vitro using the V-PLEX SARS-CoV-2 Panel 13 (IgG) kit (Meso Scale Discovery, Rockville, USA) according to the manufacturer's instructions. The assay protocol involved blocking the plates with MSD Blocker A, followed by the addition of reference standards, controls, and serial dilutions of the COVA1-27 and COVA1-27-DFO antibodies: seven threefold dilutions, starting at 1.33 μg/mL, were prepared to assess binding. After incubation, a detection antibody (MSD SULFO-TAG™ Anti-Human IgG Antibody) was added, followed by the addition of MSD GOLD™ Read Buffer B. The plates were subsequently analyzed using a MESO QuickPlex SQ 120MM Reader. The data were processed using Discovery Workbench software and recorded as electrochemiluminescent (ECL) signals. Binding efficiency was quantified as the ratio of the ECL signal of the COVA1-27-DFO to that of the COVA1-27 antibody across the antibody dilutions. The stability assay consisted of incubating the COVA1-27-DFO antibody at three different dilutions (0.05 μg/mL, 0.02 μg/mL, and 0.005 μg/mL) in macaque serum at 37 °C for up to two weeks. The sera were analyzed after 0, 1, 3, 7, and 13 days of incubation. Results are expressed as the percentage of the ratio of the ECL signal (or AU/mL) at each time point relative to the signal at day 0.

### Ethics and biosafety statement

Young adult male and female CMs (*Macaca fascicularis*, 6 males and 7 females, aged 3.19 ± 0.47 years), F1 and F2 generations originating from Mauritian AAALAC-certified breeding centers, were used in this study. All animals were exempt from previous SARS-CoV-2 infection and did not exhibit lung lesions as observed by CT before the beginning of this study.

All CMs were housed at the IDMIT infrastructure facilities (CEA, Fontenay-aux-roses) under BSL-2 and BSL-3 containment, (Animal facility authorization #D92-032-02, Préfecture des Hauts de Seine, France) and in compliance with European Directive 2010/63/EU, French regulations, and the Standards for Humane Care and Use of Laboratory Animals of the Office for Laboratory Animal Welfare (OLAW, assurance number #A5826-01, US). The protocols were approved by the institutional ethical committee 'Comité d'Ethique en Expérimentation Animale du Commissariat à l'Energie Atomique et aux Energies Alternatives' (CEtEA number 44) under statement number A20-063. The study was authorized by the "Research, Innovation and Education Ministry" under registration number APAFIS #28752-2020121710032163v1. Animals from the different experimental groups were housed in separated BSL-2/BSL-3 rooms in which several cages modules were positioned to house one single animal per module to avoid viral & radioactive contamination between individuals but to preserve eye & sound contacts between animals.

### Animals and study design

Eight CMs (4 males & 4 females) were exposed to a $1 \times 10^5$ TCID$_{50}$ dose of the SARS-CoV-2 Delta variant (lineage B.1.617.2–isolate hCoV-19/USA/PHC658/2021, BEI NR-55612) and three CMs (1 male & 2 female) to PBS. Two additional CMs (1 male & 1 female) were

convalescent animals infected by the SARS-CoV-2 Delta variant (same infectious dose) three months before the study. For exposure, animals were premedicated using atropine (0.04 mg.kg$^{-1}$) and anesthetized with ketamine (5 mg.kg$^{-1}$) and medetomidine (0.05 mg.kg$^{-1}$). For all other procedures, the same anesthesia was used without atropine. Both mock and viral exposure were performed using the intranasal (250 μL in each nostril) and intratracheal routes (4.5 mL) at day 0. Eleven macaques (4 males, 7 females) were additionally injected intravenously with either 2.5 mg ($n = 8$) or 150 μg ($n = 3$) IgG1 monoclonal human antibody COVA1-27. Two macaques (2 males) were injected intravenously with 2.5 mg human IgG1 whole molecule as a control. Both antibodies were coupled with [$^{89}$Zr] chelated by deferoxamine (DFO) as described in the previous section. The groups are described in Table 1. The intravenous injection of the radiotracers was performed one day prior to the first imaging session to allow whole-body biodistribution of the molecule.

Animals of the high-dose group, followed for 7 or 14 days ($n = 8$, cf. Table 1), were also injected subcutaneously with 25 μg (100 μL) of purified spike protein (SARS-CoV-2 wildtype, S_2P-Foldon-His, Mitch 20211122, diluted in DPBS) in the left thigh and 100 μL DPBS (ref 14190-094, Gibco) in the right thigh to evaluate recognition of the spike protein in vivo. The body weight and temperature of the macaques were recorded at each sampling time point.

PET/CT imaging was performed on days 0, 1, 2, 3, 7, 10, and 14 post-exposure (pe), corresponding to the presence of the virus during the acute phase of infection[25]. Convalescent animals were imaged with the same timing for one week after injection. During acquisition, the macaques were maintained under 0.5–2% isoflurane. CT was used to evaluate the lung lesions and locate the PET signal. Nasopharyngeal, tracheal, and rectal swabs (collected in Viral Transport Medium, 3 mL, CDC, DSR-052-01) were performed for each animal on days 1, 2, 3, 4, 8, 11, and 15 pi to assess viral titers, as well as local radioactivity. Blood was collected (1 mL, EDTA) on days 1, 2, 3, 4, 8, 11, and 15 pi to assess blood radioactivity. All sampling and imaging experiments were also performed one month before exposure to assess baseline levels. Convalescent animals were followed for one week with the same sampling and schedule.

### Quantification of radioactivity

The radioactivity was quantified for nasal, tracheal and rectal swabs, as well as whole blood and associated plasma for every sampling time described above. Briefly, samples (3 mL for swabs in VTM, 700 μL whole blood and associated plasma) were loaded into a gamma counter (Hidex Automatic Gamma Counter, Finland), weighted, and counted for 1 min (energy window 480–558 keV). Of note, plasma was obtained from the supernatants of EDTA-treated blood samples after centrifugation (1800 × *g*, 10 min). All specimens were stored at −80 °C for one month after the injection date to allow radioactive decay before establishing the viral load.

### Quantification of the virus in fluids by RT-qPCR

Genomic RNA (gRNA) and subgenomic RNA (sgRNA) from nasal and tracheal swabs (diluted in VTM) were extracted using the NucleoSpin™ 95 virus core kit (Macherey-Nagel) according to the manufacturer's protocol. Reverse transcription (RT) and quantitative polymerase chain reaction (qPCR) of gRNA and sgRNA were performed to assess the viral titers using the SuperScript III platinum one-step quantitative RT-qPCR System and RNase out (Life Technologies, Invitrogen) and CFX96 Touch Real-Time PCR Detection System (Bio-Rad). The following primers and probes were used: primers *IP4* gene (SARS-CoV-2) F GGTAACTGGTAT-GATTTCG, R CTGGTCAAGGTTAATATAGG and *IP4* gene probe FAM-TCATACAAACCACGCCAGG-BHQ1 for gRNA, primers *E* gene (SARS-CoV-2) F CGATCTCTTGTAGATCTGTTCTC, R ATATTGCAGCAGTACG CACACA and *E* gene probe HEX-ACACTAGCCATCCTTACTGCGCTTCG-

BHQ1 for sgRNA (Eurofins Genomics). The estimated lower limit of detection (LoD) was $5.79 \times 10^2$ copies/mL and $7.49 \times 10^2$ copies/mL of SARS-CoV-2 gRNA and sgRNA respectively. The estimated lower limit of quantification (LoQ) was $5.79 \times 10^3$ copies/mL and $7.49 \times 10^3$ copies/mL of SARS-CoV-2 gRNA and sgRNA, respectively.

## Quantification of the virus in tissues by RT-qPCR

All tissue samples were collected at euthanasia and stored dry in cryotubes at −80 °C after gamma counting. Tissue fragments were obtained from the lungs, trachea, heart, spleen, kidney, liver, and axillary, inguinal, and tracheobronchial lymph nodes. A fragment of 50 mg of frozen sample was lysed in NucleoZOL (Macherey-Nagel, Duren, Germany) using a Precellys Evolution/Cryolys Device (Bertin Technology, Montigny-le-Bretonneux, France). RNA isolation was performed using a NucleoSpin™ RNA Core Kit (Macherey-Nagel) according to the manufacturer's instructions. Reverse transcription and qPCR were performed using the SuperScript III Platinum One-step Quantitative RT-qPCR System (Life Technologies). The protocol used to quantify SARS-CoV-2 *IP4* genomic mRNA (gRNA) was as previously described for liquid samples, with an estimated lower LoD of $2.24 \times 10^2$ copies/mL of SARS-CoV-2 gRNA and an estimated lower LoQ of $2.24 \times 10^3$ copies/mL.

## CT and PET-CT acquisition

All image acquisition was performed using the same clinical Digital Photon Counting PET-CT system (Vereos-Ingenuity, Philips) implemented in a BSL-3 laboratory[64].

All sessions were performed using the same experimental conditions (acquisition time and animal order) to limit experimental bias. Animals were anesthetized with ketamine and medetomidine as already described, intubated, and then, maintained under 0.5–2% isoflurane. They were placed in a supine position on a warming blanket (Bear Hugger, 3 M) on the machine bed, with monitoring of the cardiac rate, oxygen saturation, and temperature.

For PET-CT acquisitions, CT was performed twice, once under breath-hold (for CT anatomical segmentation) and again prior to PET acquisition for attenuation correction and anatomical localization.

All technical parameters, detector collimation (64 × 0.6 mm), tube voltage (120 kV), and intensity (150 mA) were identical for all acquisitions. Chest-CT images were reconstructed with a slice thickness of 1.25 mm and an interval of 0.625 mm. Whole-body CT images were reconstructed with a slice thickness of 1.5 mm and an interval of 0.75 mm.

A whole-body PET scan (3–4 bed positions, 3 min/bed position) was performed 24 h and on days 1, 2, 3, 4, 8, 11, and 15 post-injection of 150 μg/animal (low dose) or 2.5 mg/animal (high dose) of [89Zr]COVA1-27-DFO or [89Zr]IgG1-DFO via the saphenous vein. PET images were reconstructed onto a 256 × 256 matrix using OSEM (3 iterations, 15 subsets).

## Image analysis

Image analysis was carried out on whole organs: brain, nasal cavity, lungs, trachea, heart, liver, spleen, kidneys, tracheobronchial, axillary, and inguinal lymph nodes using the free software 3Dslicer (v.5.0.3.). Standard uptake deviation (SUV) takes into account the radioactive decay of [89Zr]. It was used to analyze the PET signal in the abovementioned compartments. The regions of interest (ROI), brain, nasal cavity, lungs, trachea, heart, spleen, liver, and kidneys, were defined using the CT signal and corrected using the PET signal, when necessary, using the same threshold/time point. The following 3D slicer functions were used to generate the segmentations: chest imaging platform, fast marching, paint, draw, and fill between slices. All segmentations were normalized against the PET signal of the segmentation at D1 post-tracer injection. On the contrary, sub-cutaneous injection sites and lymph nodes were defined using the

PET signal with the help of the CT signal for anatomical localization. For all SARS-CoV-2 exposed animals, lesional areas in the lungs were identified using the CT signal; 3 to 4 identical spheres per lesion (volume = 0.07 ± 0.01 mL) were drawn and compared to the same ROI at the same location in the lungs of control NHPs. Similarly, the same spheres were drawn in CT-lesion-exempt areas of the lungs of all animals for the analysis of non-lesional areas of the lung. Pulmonary lesions were defined as ground-glass opacity, crazy-paving pattern, or consolidations, as previously described[65]. Two to three individuals assessed the lesion features (type (scored from 0 to 3) and extension (scored from 0 to 3)) independently for each lung lobe and the final CT score results (sum of each lung lobe score) were determined by consensus as described previously[47]. Pre-existing background lesions were scored 0.

## Statistical analysis

Data are presented as individual values with the mean and standard deviation. Linear regression parameters and Mann–Whitney ($p < 0.05$) unpaired two-sided non-parametric *t*-test results (without adjusting for multiple testing) were calculated using GraphPad® Prism 8 software. Main individual data were also reported in Supplementary table 2.

**Euthanasia at the acute phase of the infection.** Two CMs (CM9 & CM10) were injected with 2.5 mg [89Zr]COVA1-27-DFO and exposed to SARS-CoV-2 24 h after tracer injection. The radioactivity of the specimens from nasal and tracheal swabs, as well as blood and plasma, were evaluated using the same gamma counter (Hidex Automatic Gamma Counter, Finland). The biodistribution of the radiotracer was followed by PET/CT imaging in the nasal cavity, lungs, trachea, heart, liver, spleen, kidneys, inguinal, axillary, and tracheobronchial lymph nodes at 0, 1, 2, and 3 dpe. At 3 dpe, the animals were anesthetized and euthanized by intravenous injection pentobarbital (180 mg.kg⁻¹). Organs of interest were placed on the PET/CT bench and further imaged to more precisely locate the signal. The PET signal was then compared to RT-qPCR of the tissues in the ROI and gamma counts of the same tissues. The organs were fixed in 10% formalin for 24 h at room temperature and then transferred to PBS at 4 °C before being analyzed by in situ hybridization (RNAscope®) after radioactive decay.

## In situ hybridization of SARS-CoV-2 RNA in tissues

After 24 h of formalin fixation by immersion and PBS transfer, organs were included (Excelsior ES, Thermo Scientific) and embedded in paraffin. Four micron slices for each organ of interest were generated using a microtome (Leica RM2255) and mounted on SuperFrost Plus™ slides (ThermoFisher Scientific). In-situ hybridization (RNAscope®) experiments were performed using a Ventana® Ultra medical system (Roche Diagnostics) with the following probes: Probe V-nCoV2019-S-C1, S gene encoding the spike protein (Catalog No. 848561), positive control UBC (Catalog No. 310041), and negative control bacterial gene diaminopimelate (DapB) (Catalog No. 310043), as described in the literature[53]. The slides were counterstained using 0.02% ammonia water in a Leica st5020 coloring tray and mounted using Eukitt® mounting medium.

## Reporting summary

Further information on research design is available in the Nature Portfolio Reporting Summary linked to this article.

## Data availability

All quantitative data (Source data file) generated in this study have been deposited in the FigShare database (https://doi.org/10.6084/m9.figshare.28451135) under https://doi.org/10.6084/m9.figshare.28451135. Source data are provided with this paper.

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

## Acknowledgements

We thank E. Burban, E. Navarre, S. Langlois, Q. Sconosciuti, V. Magneron, M. Potier, J. M. Robert, F. Misplon, E. Sizun, and the Animalliance team for help with the animal studies, J. Morin, O. Dissake Attiapo, K. Lheureux, L. Junges, M. Galpin-Lebreau, and L. Pintore for the RT-qPCR, A-G. Garnier, C. Mayet, and J. Lemaître for help with in vivo imaging studies, J. Dinh, and E. Guyon for help with sample processing, the Fondation Bettencourt Schueller and the Region Ile-de-France for their contribution to the implementation of the imaging facilities, and the Domaine d'Intérêt Majeur (DIM) 'One Health' for its support. This study received financial support from the Agence Nationale pour la Recherche (ANR; ImaCov-Prim, ANR-20-COV8-0001-01) and France Life Imaging (TN & FC). The Infectious Disease Models and Innovative Therapies (IDMIT) research infrastructure is supported by the 'Programme Investissements d'Avenir', managed by the ANR under reference ANR-11-INBS-0008 (RLG).

## Author contributions

A.D. designed and carried out the experiments, analyzed the data, and interpreted the results. S.H., T.G., J.S., and L.B. carried out experiments, analyzed data, and interpreted results. M.J.G., F.C., and R.W.S. contributed to the study design and interpreted the results. C.C., F.R., M.C., N.D.B., Q.P., and V.C. designed the experiments, analyzed the data, and interpreted the results. R.L.G. and T.N. designed the experiments, analyzed the data, interpreted the results and supervised the study. F.C. and T.N. obtained funding. A.D., R.L.G., and T.N. wrote the manuscript. All authors critically reviewed the manuscript and contributed to the final version.

## Competing interests

The authors declare no competing interests.
