## [Transparent Peer Review file · Nature Communications]

Whole-body visualization of SARS-CoV-2 biodistribution in vivo by immunoPET imaging in non-human primates

Corresponding Author: Dr Thibaut Naninck

Version 0:

Reviewer comments:

Reviewer #2

(Remarks to the Author)

Detrille and colleagues provide data and analysis of immunoPET imaging in NHPs challenged or convalescent after exposure to SARS-CoV-2. They report finding evidence of residual antigen or infection in the lung and brain of convalescent animals as well as the nasal cavity, trachea, lungs, and kidneys of acutely challenged animals. Imaging was correlated with swab-based virology and radioactivity detection. Using a small sub-cohort of animals, they validated the imaging findings using imaging led tissue virology and pathological analysis. The work employed the use of a functionalized, non-neutralizing antibody specific to SARS-CoV-2 which had a very high binding affinity but where functionalization did not appear to impact binding. The representative images were convincing in terms of localization and explanations of how imaging in the brain was possible with a monoclonal antibody that would normally not likely cross the BBB is acceptable, especially in light of limited imaging in the acutely infected animals. Unfortunately, the work appears mostly a description and validation of the methodology and justification of its use, rather than hypothesis driven to address a specific aspect of CoVID19 biology.

Some criticism:

Justification for use of NHP vs other animal models is weak. There needs to be stronger justification in terms of using this model versus a mouse, hamster, or ferret which all show similar pathologies. A better elucidation on why NHPs are preferred here is needed.

Please include geographic origin of CMs used. This is helpful to compare against other published studies using Cynos and is common practice by investigators using NHPs.

Reviewer #3

(Remarks to the Author)

This interesting study uses a PET tracer labeled non-neutralizing antibody against the SARS CoV2 spike protein to identify where the virus resides in acutely infected and previously infected cynomolgus macaques. To my knowledge, no other such study has been done with SARS CoV2 infected macaques. Particularly interesting is the demonstration of virus in the brain of convalescent animals (but not acute animals). The strengths of this study include longitudinal analyses in acutely infected animals, validation of signal via tissues at necropsy and the ability to identify virus (or at least spike protein) in convalescent animals. The post-mortem scans with validation of radioactivity in the tissues as well as the in situ hybridization are impressive. Although understandable, the small numbers of animals and variability across animals (which mimics human variability of course) makes this a qualitative study, but there are useful takeaways from the data provided. I have relatively minor comments that might help improve the manuscript. It would also be useful to expand a bit more on the future implications of the study as well as future use of this technology in COVID research (drugs?)

1. Please explicitly state the quantitative definition of the CT score. It is not clear how this was defined or determined.
2. A table similar to Table 1 in which the data from each animal are provided would be very helpful in interpreting the results

in a summary form. For example, for each animal: how many lesions, in which tissues, total PET activity in each tissue, viral burden from swabs or other, radioactivity in tissues where determined, etc. This would be extremely helpful as it can be a bit difficult to follow in the manuscript, including keeping all the animals straight by animal number and study.

3. How many lung (or other tissue) lesions were seen by CT and also were PET+ in each animal? Representative scans were shown, which is fine, but a quantification of lesions would be useful.

4. Figure 2 Add convalescent animals to experiment timeline. i.e. viral exposure at -3 months.

5. The first line of the abstract: I assume this is confirmed cases from WHO and we know there are likely many unconfirmed cases. Perhaps the words "at least" could be used instead of "nearly", or include the word "confirmed".

6. Line 113: Tracer uptake by these 114 lesional lung regions (determined by CT) was generally higher than by non-lesional lung areas (0.28 ± 0.11 versus $115 0.21 \pm 0.10$) in these SARS-CoV-2 convalescent animals, without being significant ($p = 0.39$). This was confusing since the figure 3f shows that p value is non-lesional lung comparison of 0.39. I think there is a mistake in the text and this should be a comparison of non-lesional in convalescent vs mock.

7. Lines 169-170: We detected qualitative uptake 170 by the trachea of certain $[^{89}\text{Zr}]\text{COVA1-27-DFO}$ -injected and infected animals. Please use a better word than "certain" as this doesn't make sense. How many animals vs total animals was this detected, recognizing the difficulty in analyzing the trachea due to the intubation.

8. The "non-specific" tracer (IgG) seems very high in most of these studies (for example Figure 8f. Although somewhat addressed in the discussion, this makes the results somewhat less compelling. It is not clear why the non-specific tracer IgG is observed at such high levels, often near the level of the anti-spike antibody. Could a different non-specific antibody (i.e. anti-SIV antibody) be used to provide better distinction between COVA1-27 and IgG?

9. Although the authors provide clear information regarding the choice of antibody COVA1-27, were any others from that same reference tried in vivo to choose COVA1-27?

10. Consider combining Figures 5 and 6 since they seem like a natural fit.

11. In terms of statistics, Mann-Whitney is stated but surely the authors should have controlled for multiple comparisons or stated that this was not done.

12. lines 160- 164 and Figure 6 There is a marked change in radioactivity for all animals, but the RNA titers do not change as dramatically for 3 of the 4 animals. If it's stated that the RNA titer for CM 6 is stable, it would also be fair to explicitly state the same is true for CM 8.

Version 1:

Reviewer comments:

Reviewer #2

(Remarks to the Author)

No further comments from my end

Reviewer #3

(Remarks to the Author)

The authors have responded fully to our original concerns.

Whole-body visualization of SARS-CoV-2 biodistribution *in vivo* by immunoPET imaging in non-human primates

Alexandra Detrille¹, Steve Huvelle^{1,2}, Marit J. van Gils^{3,4}, Tatiana Geara¹, Quentin Pascal¹, Jonne Snitselaar^{3,4}, Laetitia Bossevot¹, Mariangela Cavarelli¹, Nathalie Dereuddre-Bosquet¹, Francis Relouzat¹, Vanessa Contreras¹, Catherine Chapon¹, Fabien Caillé², Rogier W. Sanders^{3,4}, Roger Le Grand¹, and Thibaut Naninck¹

Point by point responses to reviewers :

We would like to thank a lot the reviewers for their time dedicated to our manuscript evaluation. We appreciated a lot their feedback and their interest in the proposed work. Authors tried to address their comments in the revised manuscript and in this point-by-point response document.

Reviewer 1:

Detrille and colleagues provide data and analysis of immunoPET imaging in NHPs challenged or convalescent after exposure to SARS-CoV-2. They report finding evidence of residual antigen or infection in the lung and brain of convalescent animals as well as the nasal cavity, trachea, lungs, and kidneys of acutely challenged animals. Imaging was correlated with swab-based virology and radioactivity detection. Using a small sub-cohort of animals, they validated the imaging findings using imaging led tissue virology and pathological analysis. The work employed the use of a functionalized, non-neutralizing antibody specific to SARS-CoV-2 which had a very high binding affinity but where functionalization did not appear to impact binding. The representative images were convincing in terms of localization and explanations of how imaging in the brain was possible with a monoclonal antibody that would normally not likely cross the BBB is acceptable, especially in light of limited imaging in the acutely infected animals. Unfortunately, the work appears mostly a description and validation of the methodology and justification of its use, rather than hypothesis driven to address a specific aspect of COVID19 biology.

We thank the reviewer for these comments. Our work is indeed a minimally invasive methodology to track SARS-CoV-2 virus (or at least the Spike protein) longitudinally in NHPs. This methodology was however implemented in the context of long-COVID investigation in which we hypothesized that chronic inflammation observed in patients may be driven by virus and/or viral antigen persistence, especially in lungs and brain, we tried to emphasize this aspect in the introduction (line 82) and the discussion sections of the manuscript (lines 268-270)

Some criticism:

Justification for use of NHP vs other animal models is weak. There needs to be stronger justification in terms of using this model versus a mouse, hamster, or ferret which all show similar pathologies. A better elucidation on why NHPs are preferred here is needed.

We thank the reviewer for this comment and we fully agree that not enough justifications of NHP use were given in the initial manuscript. We tried to address this comment in the introduction (lines 53-60).

Please include geographic origin of CMs used. This is helpful to compare against other published studies using Cynos and is common practice by investigators using NHPs.

Thanks a lot for this comment. *Cynomolgus* macaques are all from Mauricius origin. This statement is mentioned in the methods section (line 424) in the 'Ethics and biosafety statement' paragraph.

Reviewer 2:

This interesting study uses a PET tracer labeled non-neutralizing antibody against the SARS CoV2 spike protein to identify where the virus resides in acutely infected and previously infected cynomolgus macaques. To my knowledge, no other such study has been done with SARS CoV2 infected macaques. Particularly interesting is the demonstration of virus in the brain of convalescent animals (but not acute animals). The strengths of this study include longitudinal analyses in acutely infected animals, validation of signal via tissues at necropsy and the ability to identify virus (or at least spike protein) in convalescent animals. The post-mortem scans with validation of radioactivity in the tissues as well as the in situ hybridization are impressive. Although understandable, the small numbers of animals and variability across animals (which mimics human variability of course) makes this a qualitative study, but there are useful takeaways from the data provided. I have relatively minor comments that might help improve the manuscript. It would also be useful to expand a bit more on the future implications of the study as well as future use of this technology in COVID research (drugs?)

We thank a lot the reviewer for these positive and constructive comments. Regarding future perspectives we expand a bit the implications on long-COVID research (assessment of possible viral antigen persistence inducing chronic inflammation) and on drug/vaccine evaluation in the discussion.

1. Please explicitly state the quantitative definition of the CT score. It is not clear how this was defined or determined.

We thank the reviewer for this comment and we indeed specified the CT score determination in the method section (lines 522-526).

2. A table similar to Table 1 in which the data from each animal are provided would be very helpful in interpreting the results in a summary form. For example, for each animal: how many lesions, in which tissues, total PET activity in each tissue, viral burden from swabs or other, radioactivity in tissues where determined, etc. This would be extremely helpful as it can be a bit difficult to follow in the manuscript, including keeping all the animals straight by animal number and study.

We agree with reviewer comments and added in SI 11 a table including the main data obtained for a better understanding of our results.

3. How many lung (or other tissue) lesions were seen by CT and also were PET+ in each animal? Representative scans were shown, which is fine, but a quantification of lesions would be useful.

Lesions were assessed in lungs for all animals. All infected animals showed small GGO lung lesions (with a quantification of a positive CT score of 1 or 2 at day 3 post exposure) mostly in the caudal lung lobes. Regarding PET values inside lesions, we chose to draw regions of interest inside all these lesions but also outside them to assess differences in terms of tracer uptake as described in the lung section of the manuscript. However, as COVA1-27 was described here as a novel imaging probe and we

conducted only few animals per conditions we did not define any threshold of positivity to grade 'PET+' lesions as it can be done using [¹⁸F]FDG PET in the oncology field for instance. However, quantification of global tracer uptake in lesions vs non-lesional areas of the lung is provided in figure 7.

4. Figure 2 Add convalescent animals to experiment timeline. i.e. viral exposure at -3 months.

We agree with this remark and included a second timeline for convalescent animals in figure 2.

5. The first line of the abstract: I assume this is confirmed cases from WHO and we know there are likely many unconfirmed cases. Perhaps the words "at least" could be used instead of "nearly", or include the word "confirmed".

We thank the reviewer for this comment and changed the abstract accordingly (line 18).

6. Line 113: Tracer uptake by these 114 lesional lung regions (determined by CT) was generally higher than by non-lesional lung areas (0.28 ± 0.11 versus $115 \ 0.21 \pm 0.10$) in these SARS-CoV-2 convalescent animals, without being significant ($p = 0.39$). This was confusing since the figure 3f shows that p value is non-lesional lung comparison of 0.39. I think there is a mistake in the text and this should be a comparison of non-lesional in convalescent vs mock.

We thank the reviewer for pointing out this mistake in the manuscript and apologize for it. COVA1-27 uptake in lesional and unlesional areas of convalescent animals was indeed not significant ($p=0.39$) while comparison of COVA1-27 uptake in non-lesional areas in convalescent vs mock animals was significant ($p=0.01$). We corrected the manuscript accordingly (figure 3f and main text).

7. Lines 169-170: We detected qualitative uptake 170 by the trachea of certain [⁸⁹Zr]COVA1-27-DFO-injected and infected animals. Please use a better word than "certain" as this doesn't make sense. How many animals vs total animals was this detected, recognizing the difficulty in analyzing the trachea due to the intubation.

We agree with this comment. We better specified our statement in the manuscript ('certain' was a grammatical mistake and we apologize for it) indicating that COVA1-27 signal was detected in 3 out of the 4 animals studied in this exposed+COVA condition (line 180).

8. The "non-specific" tracer (IgG) seems very high in most of these studies (for example Figure 8f. Although somewhat addressed in the discussion, this makes the results somewhat less compelling. It is not clear why the non-specific tracer IgG is observed at such high levels, often near the level of the anti-spike antibody. Could a different non-specific antibody (i.e. anti-SIV antibody) be used to provide better distinction between COVA1-27 and IgG?

This comment is actually relevant. We chose for this project to use a commercial IgG isotype control as negative control in order to be 'easily' reproducible or comparable for further studies. Part of the non specific binding is certainly due to the uptake by their FcγRs as already discussed in the discussion section. However, we cannot exclude additional non-specific binding due to off-target effect since we had no access to the sequence of the recognized epitope to assess possible partial cross-reactivity with non viral proteins, although we confirmed in this study its lack of binding to the spike protein as shown in figure 1. We mentioned in the discussion section that testing additional non-specific control antibodies may be needed to identify the causes of the observed background (lines 317-318).

9. Although the authors provide clear information regarding the choice of antibody COVA1-27, were any others from that same reference tried in vivo to choose COVA1-27?

Thanks a lot for this really relevant comment. COVA1-27 choice was made after *in vivo* testing of the neutralizing efficacy of COVA1-18 against SARS-CoV-2 *in vivo* in NHP¹. For SARS-CoV-2 imaging we then selected, among all 'COVA' mAb isolated from patients² one of the mAbs with similar affinity of COVA1-18 against the spike protein but without the neutralizing activity. We now added references of COVA antibodies line 87.

10. Consider combining Figures 5 and 6 since they seem like a natural fit.

We agree with this comment and combined these figures and changed the text accordingly.

11. In terms of statistics, Mann-Whitney is stated but surely the authors should have controlled for multiple comparisons or stated that this was not done.

We thank the reviewer for this comment and changed the statistical method accordingly (no multiple comparison performed).

12. lines 160- 164 and Figure 6 There is a marked change in radioactivity for all animals, but the RNA titers do not change as dramatically for 3 of the 4 animals. If it's stated that the RNA titer for CM 6 is stable, it would also be fair to explicitly state the same is true for CM 8.

We agree with this comment and added the comment that CM8 RNA titers in the nasopharynx remained stable between D2 & D3 while the radioactivity decreased (line 173).

- 1 Maisonasse, P. *et al.* COVA1-18 neutralizing antibody protects against SARS-CoV-2 in three preclinical models. *Nature Communications* **12**, 6097, doi:10.1038/s41467-021-26354-0 (2021).
- 2 Brouwer, P. J. M. *et al.* Potent neutralizing antibodies from COVID-19 patients define multiple targets of vulnerability. *Science* **369**, 643-650, doi:10.1126/science.abc5902 (2020).